# Deciphering and Enhancing Commonsense Reasoning in LLMs from the Perspective of Intrinsic Factual Knowledge Retrieval

## Abstract

Commonsense reasoning in large language models (LLMs) bridges the gap to physical world, thus allowing them to think and behave more like humans. Previous research has shown that LLMs acquire the underlying factual knowledge from extensive training corpora and store it within their parameters. However, how LLMs apply this knowledge during the inference phase remains unclear. This lack of transparency makes it difficult to determine whether shortcomings in LLMs are due to a lack of factual knowledge or insufficient reasoning capabilities. In this work, we aim to decipher the commonsense reasoning process into human-understandable steps. By interpreting the hidden states in different transformer layers and token positions, we uncover a specific mechanism by which LLMs execute reasoning. Our extensive experiments indicate: 1) both attention head and multi-layer perceptron (MLP) contribute to the generation of factual knowledge from different perspective. 2) The process of commonsense reasoning in LLMs involves a clear sequence of knowledge augmentation, broadcast, retrieval, reranking, and answer generation. Building on these findings, we have discovered that LLMs often contain relevant facutal knowledge but fail to retrieve the correct knowledge at top. To address this issure, we selectively fine-tuned the key heads and MLPs, resulting in notably improvements in reasoning performance in both in-domain and out-of-domain settings.

## 1 Introduction

Commonsense reasoning is a human-like ability to make presumptions about the type and essence of ordinary situations humans encounter every day (Wikipedia contributors, 2023). It is the key for human to interact with the world, and also the bridge for AI systems to reason about the world as humans (Wei et al., 2022; Talmor et al., 2022; Bhargava & Ng, 2022a). Recent Large Language Models (LLMs) have demonstrated impressive abilities in commonsense reasoning, particularly when employing the chain-of-thought technique (Wei et al., 2022; Wang et al., 2022; Saparov & He, 2022). These models can answer complex questions about world knowledge with high accuracy and even offer suggestions for everyday human activities. However, they often struggle with some basic commonsense aspects, such as reversing curses (Berglund et al., 2023), which poses challenges to users trusting their results. Therefore, understanding how models perform commonsense reasoning is vital for developing AI that is both transparent and reliable.

To unravel the commonsense reasoning capabilities of LLMs, existing studies have explored how the parameters of these models encode factual knowledge, which is derived from extensive training corpora (Akyürek et al., 2022; Li et al., 2022; Petroni et al., 2019; Roberts et al., 2020; Allen-Zhu & Li, 2023). However, the underlying mechanism of how this knowledge is applied during inference is still a mystery. This uncertainty makes it difficult to determine whether errors in commonsense reasoning stem from a lack of knowledge or from flawed understanding. For instance, if a model mistaken that *Raclette and Switzerland are unrelated*. This could either be because it lacks the knowledge that *Raclette is a Swiss dish* or because it favors the perception of *Raclette is a cheese, and cheese originates from Middle East*. Motivated by this, we aim to reverse engineer the intrinsic mechanism in LLM, and decipher the commonsense reasoning process of LLMs into steps that are understandable to humans. In this way, we can better understand why models produce certain

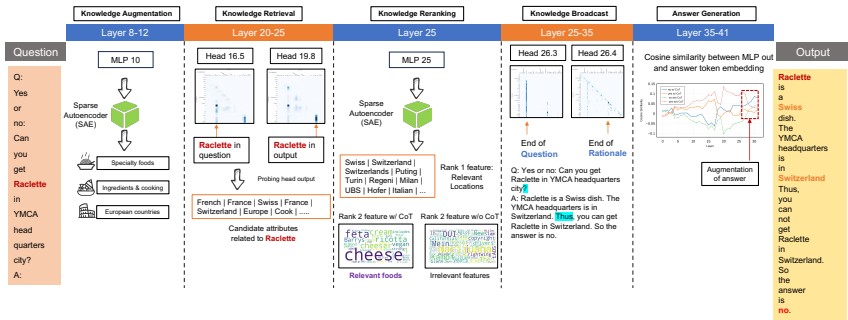

Figure 1: Deciphered commonsense reasoning process in LLMs. The five stages of the process are depicted through the example of addressing a reasoning question, as presented in the leftmost column, while the corresponding generated answer is showcased in the rightmost column, utilizing the Gemma2-9B model. The detailed depiction of these stages is presented sequentially from left to right in the central columns, corresponding to the processing order along with the associated layers. This figure is best viewed when zoomed in.

outputs or fail to generate correct answers, and we can enhance the model's reasoning capabilities in a targeted and rational manner.

In this study, we leverage a variety of analytical methods including path patching (Wang et al., 2023a), the Logit Lens (nostalgebraist, 2021), and SAE (Lieberum et al., 2024) to analyze the behavior of models from multiple dimensions. Given that commonsense reasoning is integral to the whole sequence of the rationale, our focus shifts toward examining the interrelationships between different tokens rather than delving into the details of individual token generation. To achieve this, we have designed an "Interpreting Module" that automates the analysis of how models produce individual tokens. Inspired by Bills et al. (2023) on interpreting GPT-2 using GPT-4, we also utilize GPT-4 to analyze results from Path Patching, Logit Lens, and SAE. Through comprehensive experiments, we summarized a five-stage reasoning process for factual knowledge recall, shown in Fig. 1, including knowledge augmentation, broadcast, retrieval, reranking, and finally answer generation. Specifically, LLMs first evoke related factual knowledge for augmentation. The knowledge is retained within the hidden states at each token position in the whole rationale. When predicting the key content in rationale that require commonsense reasoning, the knowledge is retrieved to provide supporting evidence. It is first recalled by attention heads and then re-ranked by multi-layer perceptrons (MLPs). At the end of rationale, the conclusion such as ··yes/no" is derived and stored in the hidden states. Finally, the answer is transferred through the heads into the output.

Building on these five stages, we identified that LLMs' failing to answer correlates with the issue of knowledge retrieval and reranking. The models misinterpret key words in the context, leading to the failure of attention heads to recall and MLPs to re-rank the correct factual knowledge at the top position. To address this problem, we fine-tuned specific heads for knowledge retrieval and MLPs for reranking, enhancing the model's ability to recall the correct knowledge, and thereby improving its reasoning performance. Experimental results demonstrate that fine-tuning less than 10% of parameters, compared to a full model fine-tuning, leads to a notable performance enhancement, especially for out-of-domain settings. This selective adjustment strategy exhibits superior performance, further validating the understanding and explaining of the reasoning process in models.

We summarize our contributions as follows: (1) We focus on interpreting the process of commonsense reasoning within LLMs into steps that are comprehensible to humans. Through experimental analysis, we found that LLMs augment related factual knowledge as a form of database, subsequently retrieving and re-ranking key tokens during prediction, and finally generating conclusions and answers. (2) Building on the above observations, we further identify that on commonsense reasoning tasks, LLMs often fail to retrieve correct knowledge, leading to erroneous reasoning or conclusions. By selectively fine-tuning key heads and MLPs, the performance of reasoning is enhanced, especially for out-of-domain samples. It validates the reliability of the interpreting results.

## 2 RELATED WORK

**Commonsense reasoning.** Machine common sense, or the ability to comprehend and reason about an open-ended world, has long been recognized as a crucial aspect of natural language understanding (Bhargava & Ng, 2022b; Sap et al., 2020). With the advent of large language models, there has been a significant leap in the reasoning capabilities of deep learning models, especially with the adoption of Chain of Thought (CoT) techniques. This has propelled the enthusiasm for understanding and advancing reasoning abilities to new heights. In this paper, we focus specifically on commonsense reasoning. Unlike temporal and numerical reasoning, which often emphasize a more symbolic approach, commonsense reasoning explores the connections between events or entities, enhancing our understanding of how large language models perceive and interpret the world.

**Large language models (LLMs).** Recent advancements in Large Language Models (LLMs) have led to remarkable performance across various Natural Language Processing (NLP) tasks. Although some commercial LLMs, such as GPT-3.5 (Brown et al., 2020) and GPT-4 (OpenAI, 2023), are closed-source, the growing number of open-source LLMs is achieving comparable results. Llama series (Touvron et al., 2023) and Gemma series (Team et al., 2024) are two families of open-source LLMs that exhibit remarkable proficiency in NLP tasks. Our experiments are conducted on four pretrained language models, Llama2-7B, Llama2-13B, Gemma2-9B, and Qwen2.5-72B (Qwen Team, 2024). The model weights for these architectures are openly accessible on HuggingFace. In performance evaluation, all these models exhibit remarkable proficiency in reasoning NLP tasks.

**Mechanistic interpretability of Large Language models.** Despite their impressive capabilities, large language models' internal mechanisms remain largely underexplored. A predominant theme is the identification of specific layers and neurons responsible for knowledge storage (Meng et al., 2022; Dai et al., 2021; Geva et al., 2023). Recent studies have introduced and refined the "path patching" approach to identify critical components in models, including GPT-2 small (0.1 billion parameters) and Chinchilla, for tasks like indirect object identification and multiple-choice questions (Wang et al., 2023b). This method, inspired by causal mediation analysis, involves perturbing component inputs and observing the resulting changes in model behavior, has been successfully extended to various tasks and larger models, demonstrating its broad applicability and scalability (Goldowsky-Dill et al., 2023; Hanna et al., 2023; Lieberum et al., 2023; Conmy et al., 2023).

A significant gap exists in LLM interpretability research, particularly in understanding the key components enabling complex tasks like reasoning. The complexity of CoT reasoning tasks makes it challenging to design a unified symbolic causal model (Geiger et al., 2023). This work uses the path patching method to identify crucial attention heads/MLPs responsible for CoT reasoning in LLMs. To validate these findings, we employ a "knockout" experiment, comparing the full model's behavior to a model without the specific head, as inspired by previous work (Wang et al., 2023b). t

## 3 METHOD

### 3.1 PRELIMINARY

In LLMs, commonsense reasoning is a multi-token generation process, including rationale and answer. Based on the construction of Subject-Verb-Object triplets (SVO) (Speer et al., 2017) used in the StrategyQA (Geva et al., 2021) and CSQA (Wikipedia contributors, 2023) datasets, we identify three key positions in the model generation: Concept, Attribute, and Response tokens. These tokens are observed special in experiments, and therefore we highlight them for better comprehension.

**Concept** ($\mathcal{C}$): The subject of inquiry in the question; this is a concept node in a knowledge graph, representing any entity, idea, or object relevant to commonsense (e.g., "Ganesha" in Figure 2.)

**Attribute** ($\mathcal{A}$): The object, which is paired with $\mathcal{C}$ as SVO to contain some knowledge, is also a concept node. These attributes, according to their relevance as accurate knowledge for the question, can be categorized into predicted attributes $\mathcal{A}_p$ (e.g., "Ganesha is a *Hindu* god") and general attributes $\mathcal{A}_g$ (e.g., "Ganesha is recognized by his *elephant* head and four arms").

**Response** ($\mathcal{R}$): The answer to the question, which can vary depending on the type of question. It may be a binary judgment (e.g., "yes/no"), a selection (e.g., "(2) Kayla"), or a free-form text.

## 3.2 INTERPRETING MODULE

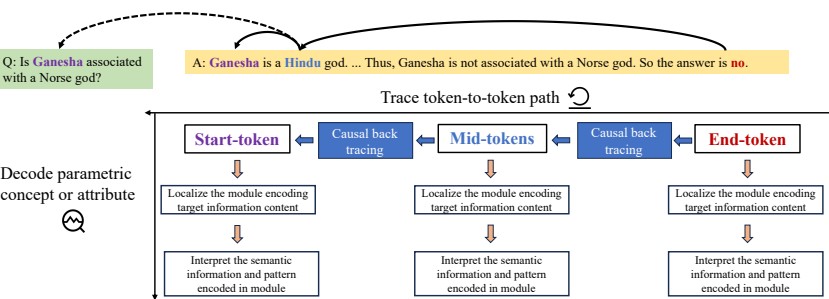

Figure 2: Overview of interpreting pipeline.

To understand how the answer is generated, we begin by extracting information from the evaluated data and the interpreted LLM, including $\mathcal{R}$, $\mathcal{C}$, $\mathcal{A}_p$ and $\mathcal{A}_g$. These tokens serve as the observed anchors to assist the understanding of the mechanism within LLMs. This extraction can be performed using GPT-4 (prompt in Appendix A.3.2) or manual methods. (see Appendix A.4 for examples.)

Since the rationale and the answer are recursively generated, it is hard to investigate both the relation across different tokens and the patterns in the token generation simultaneously. Therefore, as illustrated in Figure 2, we divide our interpretation process into two orthogonal pipelines.

**Trace token-to-token path**: The first is the horizontal pipeline, which traces the path of tokens, from the end to the start. For example, tracing from $\mathcal{R}$ to $\mathcal{A}$ then to $\mathcal{C}$. Through causal back tracing within LLMs, it reveals the relationships across the sequence of tokens.

**Decode parametric concept or attribute**: The second pipeline, shown vertically, analyzes the patterns within LLMs when generating a specific token, including inner behaviors and activation characteristics. It first identifies and localizes the modules (e.g., attention heads and MLPs) that are related to the target content (e.g., $\mathcal{R}$, $\mathcal{C}$, $\mathcal{A}_p$ and $\mathcal{A}_g$). Subsequently, it decodes the semantic information and patterns encoded in these modules into human understandable formats.

## 3.3 INSTANTIATION OF INTERPRETING MODULE

**Instantiation of tracing token-to-token path**. We employ Path Patching (Wang et al., 2023a) as an effective tool for causal back tracing. This method originates from causal mediation analysis (Vig et al., 2020), where the results of direct effect enable us to identify the significant heads. Heads with the Top-10 direct effect are considered contributors to generating a token. By examining the attention patterns in these important heads, previous tokens with attention scores greater than 0.2 are regarded to have high correlation with current token. These tokens are the targets for tracing. This process can then be iteratively applied to discover the transition path across tokens. Path Patching relies on high-quality counterfactual data, which is paired with original data to calculate the direct effect for each head. It must be carefully designed to change specific semantics within a sentence minimally, without disrupting other narrative settings. We automatically generate this counterfactual data by GPT-4 (Further details are available in Appendix A.5), achieving consistency comparable to human-generated data. (See Appendix A.3) for comparison results.)

**Instantiation of decoding parametric concept or attribute**. We use Logit Lens (nostalgebraist, 2021) to localize the modules that contain target information. This approach is able to project hidden states directly into the vocabulary space using the model's pretrained unembedding matrix. It reveals the information contained in current hidden states and explains the contribution of specific heads or MLPs or residual blocks to the predicted token. Specifically, we calculate the softmax probability of the observed tokens ($\mathcal{A}_p$, $\mathcal{A}_g$ or $\mathcal{R}$) after projecting the hidden states to vocabulary space. The probabilities across layers will form the curves (see examples in Figure 4), where layers exhibiting extreme values are identified for further analysis. For MLP, we adopt Sparse Autoencoder (SAE) (Gao et al., 2024) to decode the semantic information embedded in the parameters and activations. (e.g., Information related to "Hindu" is decoded in MLP of layer 8 when feeding "Ganesha" to the model.) Based on dictionary learning, SAE translates the internal hidden states of LLMs into several interpretable pieces, or termed latents. These latents are activated by specific token sequences, and most can be translated by GPT-4 into concrete semantic descriptions. Regarding

attention heads, we use probing to decode the semantic information. We project the outputs of the heads into the vocabulary space and examine the top-$K$ tokens in the head's output distribution to decode the semantic information. Specifically, we calculate the proportion of tokens in the top-$K$ that are correlated with the observed token.(e.g. "elephant" and "deity" are considered correlated with the observed token "Hindu"). If the proportion exceeds a pre-set threshold, we presume this head encodes concept-related attributes.

### 3.4 VERIFICATION OF INTERPRETING RESULTS

To verify the mechanism found by interpreting module, we adopt selectively supervised fine-tuning on the identified modules. Following the method proposed in Zhang et al. (2024), we directly use the same settings without modification for effective verifications. Given a sequence of attention heads ordered by their significance, denoted as $(l_1, h_1), (l_2, h_2), \ldots$, where $l_i$ represents the layer index and $h_i$ represents the head index of the $i^{th}$ ranked head, only top $K$ heads are exclusively updated during fine-tuning. We optimize both the corresponding input mapping matrix $\{W_{l_1}^{h_1}, W_{l_2}^{h_2}, ..., W_{l_K}^{h_K}\}$ and the output mapping matrix $\{O_{l_1}^{h_1}, O_{l_2}^{h_1}, ..., O_{l_K}^{h_K}\}$ in top $K$ heads simultaneously. For the selected MLP layer, we update all parameters in this layer.

## 4 EXPERIMENTS

### 4.1 EXPERIMENTS OVERVIEW

As presented in Section 3.2, we start from the end token position (i.e., the position of Response $\mathcal{R}$). At the position of $\mathcal{R}$, we decode the parametric concept during response generation and causally trace back to the previous token position (i.e., the position of attribute $\mathcal{A}$). We term this process as **answer generation**(§4.2). Trace back to the position of $\mathcal{A}$, where an analysis of the predicted attribute's generation revealed the mechanisms of **knowledge retrieval** and **reranking** (§4.3). Further tracing the source of attribute information led to the position of concept $\mathcal{C}$, uncovering the mechanisms of **knowledge augmentation** and **knowledge broadcast**(§4.4). After interpreting the mechanism behind commonsense reasoning, we employed SSFT (§4.6) to validate the mechanism.

**Models**  To explore the internal mechanisms of large language models (LLMs), we conducted experiments on open-source models, selecting diverse architectures and sizes to ensure the robustness and generalizability of our findings. Specifically, we employed Gemma2-9B (Team et al., 2024), Llama2-7B (Touvron et al., 2023), and Qwen2.5-72B (Qwen Team, 2024) The results in the Section 4 primarily focus on Gemma2-9B, as Sparse Autoencoders (SAEs) have been trained for all its layers (including residual and MLP layers) (Lieberum et al., 2024), enabling comprehensive validation of our analyses. Additional results for Llama2-7B and Qwen2.5-72B are provided in Appendix A.8 and Appendix A.7, respectively.

**Datasets**  Commonsense reasoning is inherently abstract, encompassing diverse question types and linguistic expressions. To explore the factual knowledge recall mechanism of large language models (LLMs), we selected four widely used commonsense reasoning benchmark datasets: **StrategyQA** (Geva et al., 2021), **CommonsenseQA** (Talmor et al., 2018), **WinoGrande** (Sakaguchi et al., 2021), and **SocialIQA** (Sap et al., 2019). The results are primarily reported on the StrategyQA dataset, with results for the other three datasets provided in Appendix A.6. All metrics and curves are averaged over 100 samples. Few-shot prompts from Wei et al. (2022) and Li et al. (2024) are adopted to elicit model's reasoning abilities.

### 4.2 ANSWER GENERATION

Considering examples from StrategyQA where the response $\mathcal{R}$ is "yes" or "no", we decode the correct answer and incorrect answer information in MLP outputs. As shown in Figure 3 (a), the curve of MLP in layers 0–33 contain almost no information related to the $\mathcal{R}$. However, in layers 34 and 37, the probability of the correct response exhibits a sharp increase, with two distinct spikes, while the probability of incorrect response remains unchanged. Similarly, we analyzed the attention curve (Figure 3 (b)) and found that in layers 0–31, there is minimal response-related information. However, in layers 32–35, the probabilities of both correct and incorrect responses increase significantly and

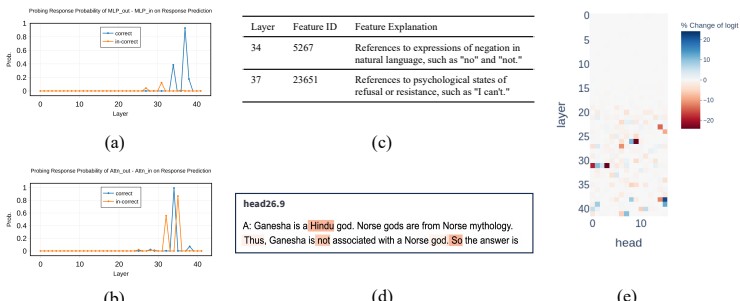

(a)             (c)

(b)           (d)           (e)

Figure 3: Probing result at answer generation position: (a) Probabilities of correct and incorrect answer of MLP outputs across layers, showing dominant information of correct answer in deep layers. (b) Probabilities from attention layers show entangled information of different types of answers in the output. (c) Information decoded in the key MLP is directly related to the correct answer. (d) Key attention head for answer generation mainly attends to the position of $\mathcal{A}$. (e) Key attention heads for answer generation are located in layers 25-37. These findings highlight the answer generation process, where head aggregate options and MLP select the answer to output.

are relatively close in magnitude. Based on these observations, we conclude that during response generation, the attention mechanism is responsible for aggregating all plausible answer options, while the MLP ultimately selects the final response to output.

Then we used Sparse Autoencoder (SAE) to analyze the information encoded in layers $34, 37$. As for the sample of "Ganesha is a Hindu god ..." with correct response as "no", we discovered numerous latents related to negation. Specifically, in layer $34$, we identified a latent corresponding to *References to expressions of negation in natural language, such as "no" and "not"*, shown in Figure 3 (c). These findings provide additional evidence supporting the critical role of the MLP in the answer generation process.

At last, we identified the key attention heads responsible for generating the conclusion and traced their information sources. These heads are concentrated in layers 25–37 (Figure 3 (e)) and primarily focus on the position of $\mathcal{A}$ (e.g. "Hindu") within the rationale (Figure 3 (d)). Despite the primary focus, we also observed some minor attentions concentrated on reasoning-related tokens (e.g., "thus" and "so"). We probe these positions through Logit Lens and found they already contain information about the correct answer (i.e., "no"). In addition, back tracing these reasoning-related tokens, the primary focuses are also "Hindu". Therefore, our investigation continuously traces back to the position of $\mathcal{A}$ prediction.

## 4.3 KNOWLEDGE RERANK AND RETRIEVAL

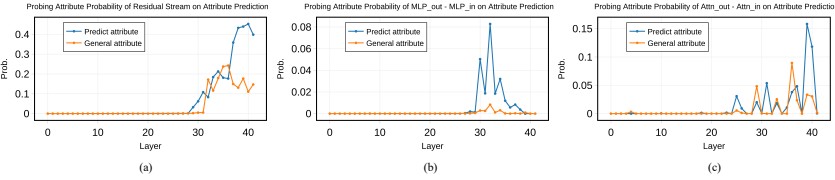

(a)           (b)           (c)

Figure 4: Probing results at the attribute prediction position: (a) Probabilities of the predicted attribute $\mathcal{A}_p$ and general attributes $\mathcal{A}_g$ of residual block outputs across layers, showing an alternating pattern in their relative importance in layers 28–40. (b) Probabilities from MLP outputs, primarily aligned with $\mathcal{A}_p$. (c) Probabilities from attention outputs, contain both $\mathcal{A}_p$ and $\mathcal{A}_g$. These results highlight the reranking mechanism, where MLP layers in the mid-to-late stages (28–38) dominate attribute selection.

The attribute information $\mathcal{A}$ decoded in residual block outputs, MLP outputs and attention outputs are shown in Figure 4. In residual block curve, the attribute information begins to emerge at around layer 30. However, the predicted attribute $\mathcal{A}_p$ is not dominant in the first place, as the probabilities of $\mathcal{A}_p$ and general attribute $\mathcal{A}_g$ increase alternately, with $\mathcal{A}_p$ gradually surpassing $\mathcal{A}_g$ at around layer 37. In MLP curve, attribute information is only evident in layers 30–37 (with probabilities close to 0 in the rest of the layers). Within these layers, it's clearly observed that $\mathcal{A}_p$ is prominently

represented, while $\mathcal{A}_g$ remains minimal. For the attention curve, in layers 25–40, $\mathcal{A}_p$ and $\mathcal{A}_g$ starts to interleave, showing no explicit dominance in between.

From the observations above, we can conclude a key finding: the MLP is responsible for enhancing the probability of $\mathcal{A}_p$ (which we termed **knowledge reranking**.) and finally generating $\mathcal{A}_p$ in attribute prediction. To validate our finding, we look into the MLP output using SAE. Specifically, we examine the features whose explanations are semantically related to both $\mathcal{A}_p$ and $\mathcal{A}_g$. The results are shown in Figure 5 (a). These features strongly represent Hindu-related attributes, but none of which is related to the general attribute $\mathcal{A}_g$. This further verifies our finding that the MLP contributes to answer generation by amplifying $\mathcal{A}_p$ only.

| Latent ID | Latent Explanation |
|---|---|
| 31.1376 | references to Indian institutions or entities |
| 31.3694 | references to Indian cuisine and food-related terms |
| 32.127330 | references to divine entities or Hindu deities |

(a)

| Attention head | Top10 tokens in vocabulary space |
|---|---|
| 25.1 | elephant, Ganes, elephant, 🐘, Elephant, Elephant, elefante, religione, estatua, Elephants |
| 25.2 | India, India, INDIA, Hindu, Indian, india, Indians, Hindus, Hindu, Mumbai |
| 29.14 | elephants, elephant, Elephants, Elephant, Elephant, elef, elephant, Ganesh, 🐘, Ganes |

(b)

Figure 5: Comparison of the information encoded in the key MLP (a) and attention heads (b) responsible for knowledge retrieval. It is observed that the MLP outputs are directly related to the final predicted attribute, whereas the attention head outputs contain various attributes associated with the concept.

Next, to understand what happens in the intertwined emergence of $\mathcal{A}_p$ and $\mathcal{A}_g$ in attention outputs, we conduct further analysis with head localization and probing. We first confirm that the most influential attention heads are localized after layer 25. The outputs of these heads encode a rich set of attribute information relevant to the concept (e.g., *elephant* and *Hindu* in the context of *Ganesha* as shown in Figure 5 (b)). Given that these attention heads operate earlier than the layers where information of $\mathcal{A}_p$ appears in the MLP (layer 30), we propose the following attribute prediction mechanism: attention heads in the intermediate layers first aggregate all relevant attributes (both $\mathcal{A}_p$ and $\mathcal{A}_g$) through a process of termed **knowledge retrieval**. Subsequently, the MLP ranks these attributes according to their relevance and selects $\mathcal{A}_p$ for the final output (i.e., **knowledge reranking**).

| Attention head | Attention score |
|---|---|
| 25.1 | Q: Yes or no: Is Ganesha associated with a Norse god?\<newline\> A: Ganesha is a |
| 25.2 | Q: Yes or no: Is Ganesha associated with a Norse god?\<newline\> A: Ganesha is a |
| 29.14 | Q: Yes or no: Is Ganesha associated with a Norse god?\<newline\> A: Ganesha is a |

Figure 6: Heads pattern for knowledge retrieval in Gemma2-9B: mainly attends to the position of concept and question end.

Finally, we find these attention heads focus on two critical token positions, as shown in Figure 6: the position of $\mathcal{C}$ and the position of question end. For example, head 25.1 exhibits average attention scores of 0.62 and 0.22 at the position of $\mathcal{C}$ and question end, respectively. Therefore, we trace back to the position of $\mathcal{C}$ to investigate the origin of $\mathcal{A}$.

## 4.4 Knowledge Augmentation and Broadcasting

From the position of the $\mathcal{A}$, we further back-trace to the positions of the $\mathcal{C}$ and the Question End. Generally, in commonsense reasoning datasets, the $\mathcal{C}$ always appears in both the question and the rationale. Through analysis, we observe that the $\mathcal{C}$ in the rationale can also back-traced to the $\mathcal{C}$ in the question. Therefore, we treat the position of $\mathcal{C}$ in the question as a focal point for deeper analysis.

Figure 7 illustrates the information curves decoded in the outputs of residual block, MLP and attention during the generation of $\mathcal{C}$, relative to the predicted attributes $\mathcal{A}_p$ and general attributes $\mathcal{A}_g$. Notably, we observe that: 1) In residual curve, it contains obvious information regarding both $\mathcal{A}_p$ and $\mathcal{A}_g$ across various layers, with $\mathcal{A}_g$ being more prominent than $\mathcal{A}_p$ at the end. 2) another two curves show that both MLPs and attention heads have large influence on $\mathcal{A}_g$ and $\mathcal{A}_p$. To further

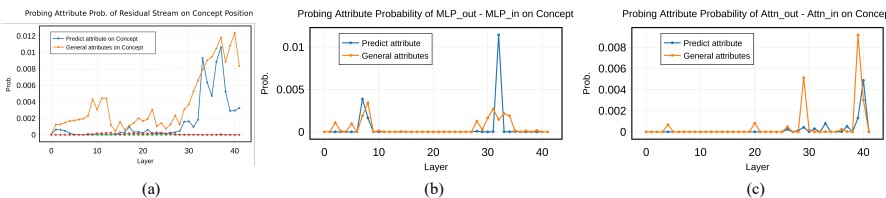

Figure 7: Decoding results of residual block (a), MLP (b), and attention (c) at the position of concept $\mathcal{C}$. Corresponding to the knowledge augmentation process: the attribute information in the shallow layer is encoded by MLP layers, which serve as the source information for knowledge retrieval.

| Layer | Feature ID | Feature Explanation |
|---|---|---|
| 7 | 119958 | References to deities and mythological figures associated with nature, fertility, and seasonal changes. |
| | 84677 | References to deities and divine entities in a religious context. |
| 32 | 109559 | References to Hindu deities and their attributes. |
| | 15523 | References to countries and regions in South Asia, particularly related to India and its cultural aspects. |

(a)

| Attention head | Top10 tokens in vocabulary space |
|---|---|
| 29.14 | elephants, elephant, elef, Elephants, Elephant, elephante, 🐘, Ganes, Gnesh, elephant |
| 29.15 | Hindu, avoent, Mharashtra, Hindu, étoient, Sri, Marathi, Sanskrit, Indian, Tamil |
| 39.7 | Lord, Krishna, lord, Ganes, LORD, Lakshmi, Krishna, Indra, Vishnu, Hindu |

(b)

Figure 8: Comparison of the information encoded in the key MLP (a) and heads (b) responsible for knowledge retrieval. It is observed that the MLP outputs are directly related to the final predicted attribute, whereas the attention head outputs contain various concept-related attributes.

validate the decoded information, as shown in Figure 8, we use SAE and Probing for investigation. Specifically, SAE identifies that MLPs in layers 7 and 32 identify latents related to "*references to Hindu deities and their attributes*". Meanwhile, Probing also reveals that heads in layers 29 and 39 rank the $\mathcal{A}_g$ at top. In addition to diminishing the impact of the information from any previous token, we also examine the three corresponding curves at the position before $\mathcal{C}$ (for instance, "Question: $\boxed{\text{Is}}$ Ganesha"). The results reveal that the information regarding $\mathcal{A}_p$ and $\mathcal{A}_g$ is virtually zero. It indicates that the emergence of $\mathcal{A}_p$ and $\mathcal{A}_g$ is indeed contingent upon the appearance of $\mathcal{C}$ and is independent of any previous tokens. In conclusion, both the MLP and heads play essential roles in assisting the model to associate and extend from $\mathcal{C}$ to related $\mathcal{A}_p$ and $\mathcal{A}_g$. We refer to this stage, along with the contributions of the MLP and heads, as **knowledge augmentation**.

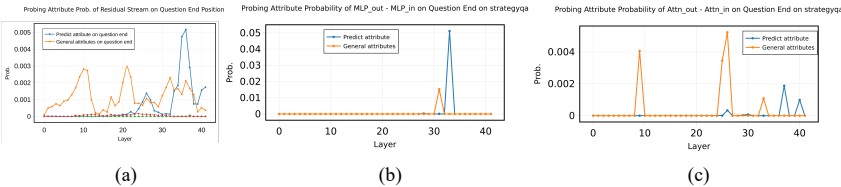

Figure 9: Decoding results of the residual block (a), MLP (b), and attention (c) at the position of the question end. Knowledge movement are discovered based on the decoding result: question end position encodes rich attribute information, which is transported by the attention. MLP adjusts the ranking of $\mathcal{A}_g$ and $\mathcal{A}_p$.

Regarding the the question's end token position, Figure 9 also presents the three corresponding curves. (1) In the residual curve, both $\mathcal{A}_p$ and $\mathcal{A}_g$ appear across multiple layers. On the contrary to concept token position, $\mathcal{A}_p$ has a greater presence than $\mathcal{A}_g$. (2) The curves for the MLP and heads also encapsulate information about both $\mathcal{A}_p$ and $\mathcal{A}_g$, and further enhance the importance of $\mathcal{A}_p$. These observations indicate that even at seemingly unrelated token positions, the $\mathcal{A}$ corresponding to the $\mathcal{C}$ (or the knowledge they encompass) can be broadcast. The original order of $\mathcal{A}$ may shift based on the current context, ultimately influencing the generation of $\mathcal{A}_p$. We term this stage as **knowledge broadcasting**.

## 4.5 SUMMARY OF THE COMMONSENSE REASONING MECHANISM

Based on the findings from the preceding subsections, we can summarize the complete mechanism of factual knowledge recall in commonsense reasoning tasks as follows:

(1)**Knowledge Augmentation**: At the position of $\mathcal{C}$, shallow-layer MLPs encode all kinds of concept-related attribute information into the residual stream. (2) **Knowledge Broadcasting**: At the question's end token position, the model aggregates information from $\mathcal{C}$ and reorganizes it based on the context. After that, the information at question's end token and $\mathcal{C}$ are broadcast to following important tokens. (3) **Knowledge Retrieval**: At the position of $\mathcal{A}$, attention heads gather attribute information from the position of $\mathcal{C}$ and the question end position. Then the information is transported to the attribute prediction position. (4) **Knowledge Reranking**: Also at the position of $\mathcal{A}$, MLP layers select the most appropriate attribute $\mathcal{A}_p$ from the retrieved candidates for prediction output. (5) **Answer Generation**: At the predict token position, attention layers aggregate information from $\mathcal{A}$ token position to draw the final conclusion.

We conducted experiments on three additional commonsense reasoning datasets (CommonsenseQA, WinoGrande, and SocialIQA) and validated the mechanisms of knowledge retrieval and knowledge reranking across all of them. However, the phenomenon of knowledge augmentation was not prominently observed in the SocialIQA and CommonsenseQA datasets. We hypothesize that this is due to the explicit provision of the required knowledge within the question context, which diminishes the model's need to infer additional related knowledge. Please see Appendix A.6 for details. In addition, we validated the proposed reasoning process on the Qwen2.5-72B model. Detailed results can be found in Appendix A.7.

## 4.6 Selective Supervised Fine-tuning on Commonsense Reasoning-related Components

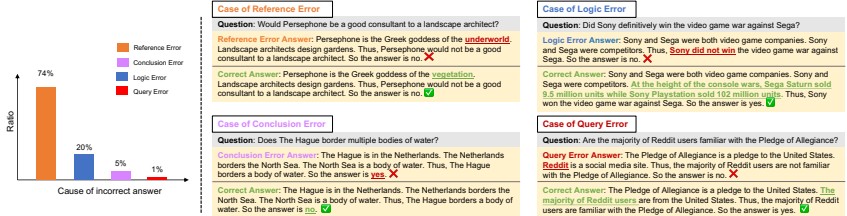

Figure 10: The distribution of the four types of errors encountered by Llama2-7B on StrategyQA. 1) Reference Error: The model retrieves irrelevant or wrong attributes. 2) Logic Error: incomplete reasoning steps. 3) Conclusion Error: reaches an incorrect answer but based on correct rationale. 4) Concept Error: incorrectly identifies the target concept for analysis. The prompt we use GPT-4 to assist classification is available in Appendix A.3.3.

Table 1: We fine-tune Llama2-7B/13B on the StrategyQA dataset using supervised fine-tuning (SFT) and selectively supervised fine-tuning (SSFT). Here are the capabilities of models on four commonsense reasoning tasks (*e.g.*, StrategyQA, CSQA, Winogrande, and SocialIQA) before and after tuning.

| | | ID Task | | OOD Task | | | | | | | | |
| | | StrategyQA | | CSQA | | Winogrande | | SocialIQA | | Average | |
| Models | Tuned Params. | Acc. | Δ | Acc. | Δ | Acc. | Δ | Acc. | Δ | Acc. | Δ |
|---|---|---|---|---|---|---|---|---|---|---|---|
| Llama2-7B | - | 62.5 | - | 61.1 | - | 53.4 | - | 60.2 | - | 58.2 | - |
| + SFT | 6.7B | 77.3 | +14.8 | 54.8 | -6.3 | 52.7 | -0.7 | 59.0 | -1.2 | 55.5 | -2.7 |
| + SSFT | 0.2B | 78.5 | +16.0 | 64.1 | +3.0 | 61.1 | +7.7 | 63.2 | +3.1 | 62.8 | +4.6 |
| Llama2-13B | - | 66.0 | - | 68.3 | - | 55.5 | - | 67.9 | - | 63.9 | - |
| + SFT | 13B | 79.0 | +13.0 | 69.5 | +1.2 | 54.6 | -0.9 | 63.1 | -4.8 | 62.4 | -1.5 |
| + SSFT | 0.5B | 80.3 | +14.3 | 72.6 | +4.3 | 56.6 | +1.1 | 69.2 | +1.3 | 66.1 | +2.2 |

**Failure Case Analysis.** By examining instances where the model (Llama2-7B) provides incorrect responses, we identified four distinct error types on the StrategyQA task (specific cases for each type are shown in Fig. 10): 1) Reference Error: The model retrieves attributes that are irrelevant to the question context or erroneous attributes; 2) Logic Error: The knowledge generated by the model

is insufficient to support the conclusions drawn by the model; 3) Conclusion Error: The model reaches an incorrect answer despite having the appropriate reasoning knowledge; and 4) Concept Error: The model incorrectly identifies the target concept for analysis. Among these, the most prevalent error type is Reference Error (74%). Furthermore, we conducted probing to investigate the underlying causes of Reference Errors, specifically determining whether these errors resulted from incorrect reranking despite the model possessing the correct attributes, or from the model's lack of knowledge regarding the correct attributes. Experimental findings indicate that the majority of Reference Errors stem from reranking issues, as the correct knowledge is typically present within the model's top five predicted tokens. Consequently, following Zhang et al. (2024) and Chen et al. (2024), we aim to enhance the model's commonsense reasoning capabilities by strategically training the identified MLP and attention heads that contribute to completing commonsense reasoning tasks, thereby improving the model's ability to recall correct attributes.

**Experiment Setup.** With the key attention heads and MLPs identified for generating attributes (refer to Section A.8 for details), we conduct the selective supervised fine-tuning (SSFT) experiment on StrategyQA task through only updating the parameter of selected heads and MLPs. Specifically, Following Fu et al. (2023) and Huang et al. (2022), each sample in our training data is organized with the format of "`{Few-shot CoT prompt}` Q: `{Question}`. A: `{Rationale}` `{Answer}`".

We selectively fine-tune the top $k$ attention heads (for knowledge retrieval) and top $l$ MLP layers (for knowledge reranking) with a learning rate of $1 \times 10^{-4}$ and a batch size of 32 for 2 epochs[1]. For supervised fine-tuning, a learning rate of $1 \times 10^{-5}$ is utilized, while all other configurations remain consistent with SSFT training. Experiments are conducted on 8 NVIDIA A100 (80GB) GPUs.

**Experiment Results.** The comparative results between SSFT and SFT are presented in Table 1. For the experiments of Llama2-7B on StrategyQA, both SSFT and SFT improved performance, achieving gains of +16.0% and +14.8%, respectively. While SFT shows a comparable enhancement for the StrategyQA task, it adversely affected performance on OOD tasks, with an average decrease of −2.7%. In contrast, SSFT continued to bolster the model's reasoning ability across all OOD commonsense reasoning tasks, improving the performance by an average of +4.6%. These findings suggest that selectively fine-tuning a small fraction of key components of LLMs on commonsense reasoning can substantially boost performance on CoT tasks (in-domain) while maintaining generalizability (out-of-domain), highlighting the effectiveness of our previous exploration. A similar trend was observed in the Llama2-13B results. Through mechanism analysis of the model before and after SSFT, we further validate that SSFT enhances the model's knowledge retrieval and reranking capabilities. (See Figure 16 for details. ). Additionally, we further validate the effectiveness of SSFT through training on three other datasets (Figure 16) and training on a larger model (Qwen2.5-72B in Table 10).

## 5 CONCLUSION

In conclusion, our research sheds light on the intricate dynamics of commonsense reasoning within LLMs, revealing a structured process that parallels human cognitive reasoning. By meticulously analyzing the hidden states across various transformer layers and token positions, we identified a multi-faceted mechanism that integrates knowledge augmentation, retrieval, and answer generation—essentially resembling a retrieval-augmented generation framework. Our findings underscore the pivotal roles played by both attention heads and MLPs in the manifestation of factual knowledge, highlighting a dual approach to knowledge processing. Furthermore, our experiments demonstrated that while LLMs often possess relevant factual knowledge, they frequently struggle to retrieve the correct information during inference. Through selective fine-tuning of key components, we achieved notable enhancements in reasoning performance across diverse contexts, indicating that targeted adjustments can effectively optimize the reasoning capabilities of LLMs. This study not only contributes to a deeper understanding of LLM functionality but also offers actionable insights for improving their reasoning processes, paving the way for more sophisticated and human-like interactions with artificial intelligence systems.

---

[1] $k = 32, l = 2$ for Llama2-7B and $k = 64, l = 2$ for Llama2-13B

## 6 ETHICS STATEMENT

This paper presents work whose goal is to advance the field of mechanistic interpretability in LLMs. We use public natural language processing datasets and leverage open-source large language models for our experiments. Our code or method are not inherently subject to concerns of discrimination/bias/fairness, inappropriate potential applications, impact, privacy and security issues, legal compliance, research integrity or research practice issues.

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

# A  APPENDIX

## A.1  COMPARISON WITH PRIOR INTERPRETING FACTUAL KNOWLEDGE WITHIN LLMS STUDIES

Our work builds upon and extends existing studies on the interpretability of factual knowledge in large language models (LLMs), distinguishing itself in terms of the reasoning process, interpreting tools, and key findings. Previous studies, such as Geva et al. (2023); Meng et al. (2022), primarily focus on a reasoning process comprising knowledge augmentation and retrieval. These works employ tools like logit lens, knockout, and causal tracing to demonstrate that factual knowledge is stored in mid-layer MLPs and retrieved by attention heads. In contrast, our study introduces a novel reasoning step, **knowledge reranking**, which highlights how deep-layer MLPs refine retrieved information to prioritize relevant attributes for final predictions.

Furthermore, while other studies such as He et al. (2024); Yuksekgonul et al. (2023) focus on distinguishing factual activation patterns or analyzing attention for entity retrieval, they do not provide a comprehensive multi-stage explanation of the reasoning process. Similarly, Yu & Ananiadou (2024) identifies knowledge storage in both attention heads and MLPs but lacks a discussion of how knowledge is effectively utilized in downstream tasks.

In addition to the reasoning process, our work advances the toolkit for interpretability research by developing and applying novel tools like path patching and sparse autoencoder (SAE). These tools enable fine-grained, token-by-token analysis of realistically queried sentences, whereas prior studies often rely on template-based triplets and tools like logit lens alone. This methodological shift allows us to uncover mechanisms such as *knowledge augmentation, retrieval, and re-ranking* in a unified framework.

## A.2  DETAILS OF INTERPRETING MODULES

| Interpreting Module | Input | Output | Conclusion (Example) |
|---|---|---|---|
| **Path Patching** | Commonsense reasoning data + counterfactual data | Distribution of head importance | For example, with "Ganesha is a Hindu god," at the "a" token position, path patching reveals which attention heads in the LLM are critical for predicting "Hindu." |
| **Logit Lens** | Commonsense reasoning data + probing attributes | Attribution of investigated attribute token within different modules (MLP, attention, residual block) | For example, with "Ganesha is a Hindu god," at the "a" token position, this method shows which attention layers transport related attributes (knowledge retrieval), and which MLP layers perform re-ranking to generate the predicted attribute. |
| **SAE** | Commonsense reasoning data + specific layer ID | Identifies what information is encoded in the output of a specific MLP layer | For example, with "Ganesha is a Hindu god," at the "Ganesha" token position, SAE helps decompose the MLP output to determine what attribute-related information is encoded. |
| **Head Pattern Analysis** | Commonsense reasoning data + specific head ID | Attention score and projection of head outputs into vocabulary space | For the example "Ganesha is a Hindu god," using the "attention head pattern analysis module" at the "is" token position, we analyzed the heads identified by path patching. This allows us to determine which important heads transported information, from which tokens, and what specific information was transported. |

Table 2: Illustration of four interpreting modules with the input, output and examples.

## A.3 GPT-ASSISTED ANALYSIS

### A.3.1 COMPARE GPT-4 WITH HUMAN

We conduct experiments to compare the "GPT-4" and "human validation" results. In the paper, GPT-4 is applied in: (1) Generation of $X_c$ (counterfactual data) using GPT-4; (2) Generation of the analysis of key component behavior using GPT-4; (3) Identification of critical position (i.e. concept, attribute, and response).

For all the scenarios, we engaged ten master's students specializing in Natural Language Processing as volunteers. Five students were manually executing all procedures, including generating $X_c$, analyzing key component behaviors, and developing data templates. The remaining students then compared their annotations with those generated by GPT-4 to judge which more accurately represented the component behavior. Overall, the results demonstrate that GPT-4 is highly accepted by human evaluators, with the combination of "GPT wins" and "Ties" exceeding $80\%$, underscoring its robust reliability. These indicate that *GPT-4's outputs are almost consistent with those generated by humans.*

Table 3: Comparison of differences between GPT-4 and human annotations

| Scenarios | GPT Wins | Human Wins | Ties |
|---|---|---|---|
| Generation of $X_c$ using GPT-4 | 8% | 12% | 80% |
| Analysis of key component behavior using GPT-4 | 12% | 10% | 78% |
| Critical position identification | 7% | 18% | 75% |

### A.3.2 PROMPT FOR POSITION EXTRACTION

We use the following prompt to assist in automatically extracting the concept, attribute, and response from the model's reasoning output.

## Prompt Template for Extraction

```
<Inputs>
{$REASONING_TEXT}
</Inputs>

<Instructions Structure>
1. Present the reasoning process text to the AI as input, labeled
    clearly with <reasoning_text> tags.
2. Direct the AI to identify and extract three key components from
    the text:
   - The "concept": The main subject or entity discussed in the
   reasoning process.
   - The "attribute": The characteristic, classification, or
   property associated with the concept.
   - The "answer": The final conclusion or decision reached
   in the reasoning process.
3. Ensure the AI outputs the result in JSON format with specific
    keys ("concept," "attribute," and "answer").
4. Include examples for clarity.
</Instructions Structure>

<Instructions>
You are tasked with analyzing a reasoning process presented in
a textual format to extract specific elements and present them
in a structured JSON output.

Here is the reasoning process text you need to analyze:

<reasoning_text>
{$REASONING_TEXT}
</reasoning_text>

Follow these steps to complete the task:
1. Identify the **concept**: The primary subject or entity discussed
    in the reasoning process.
2. Identify the **attribute**: The characteristic, classification, or
    property associated with the concept.
3. Identify the **answer**: The final conclusion or decision reached,
    typically stated explicitly in the text.

Output the results in the following JSON format:
```json
{"concept": "<concept>", "attribute": "<attribute>",
"answer": "<answer>"}
```
### Example:
Input Reasoning Process:
```

Ganesha is a Hindu god. Norse gods are associated with Norse
mythology. Thus, Ganesha is not associated with a Norse god.
So the answer is no.
```
```

**Prompt Template for Extraction**

```
Output:
```json
{"concept": "Ganesha", "attribute": "Hindu god", "answer": "no"}
```

If any of the elements (concept, attribute, or answer) are unclear
or missing from the reasoning text, leave the corresponding value
blank in the JSON output (e.g., "concept": ""). Think carefully about
the text's structure to ensure accurate extraction of each component.

Write your JSON output immediately after analyzing the reasoning
process. Do not include additional explanations or commentary.
```

### A.3.3 PROMPT FOR FAILURE CASE CLASSIFICATION

**Prompt Template for Failure Case Classification**

```
{"concept": "Ganesha", "attribute": "Hindu god", "answer": "no"}
I am testing the accuracy of a large language model's responses
on the multi-hop reasoning dataset, StrategyQA. Your task is to
classify the errors in the model's answers based on specific
error types. For each question, I will provide the input question,
the model's answer, the correct answer and the reasoning steps
needed for the correct answer. Your goal is to accurately classify
the errors using the following four error types:

1. **Entity Selection Error**: This occurs when the model picks
the wrong entity from the input, leading to incorrect reasoning
in subsequent steps.
# Example 1:
Input:
```json
{
    "question": "Are the majority of Reddit users familiar with
    the Pledge of Allegiance?",
    "model_answer": "The Pledge of Allegiance is a pledge to the
    United States. Reddit is a social media site. Thus,
    the majority of Reddit users are not familiar with the
    Pledge of Allegiance. So the answer is no.",
    "correct_answer": "yes",
    "decomposition": [
        "What country do most Reddit users come from?",
        "What country is the Pledge of Allegiance associated with?",
        "Is #1 the same as #2?"
    ]
}
```
```

## Prompt Template for Failure Case Classification

```
Classification: {"type": "Entity Selection Error", "explanation":
"The model incorrectly selected Reddit as the entity
it spoke about, while the correct entity for reasoning
should be 'Reddit users.' Therefore, this question should
be classified as an 'Entity Selection Error'".}

2. **Knowledge Retrieval Error**: This occurs when the model
retrieves irrelevant, incomplete, or incorrect knowledge,
leading to flawed conclusions in the reasoning process.
# Example 1:
...

# Example 2:
...

3. **Conclusion Misalignment Error**: This occurs when the
model's reasoning steps are correct, but the final
conclusion is wrong.
# Example 1:
...

4. **Reasoning Logic Error**: This occurs when the logical
connection between the reasoning steps and the final
conclusion breaks down. In this error, even if individual
reasoning steps are correct, they fail to coherently lead
to the intended conclusion, causing the reasoning process
to result in an illogical or incorrect outcome.
# Example 1:
...

Instructions: If the error does not fit into any of these
four categories, please suggest a new category with a clear
explanation.

For each input, I will provide the question, the model's answer,
the correct answer, and the decomposition of reasoning steps.
You should return your classification and a brief explanation as
follows:
```json
{"type": "Entity Selection Error" or "Knowledge Retrieval Error"
or "Conclusion Misalignment Error" or "Incomplete Reasoning Error",
"explanation": "Explain why this question belongs to the chosen
category."}
```
Classficiation:
```

## A.4    EXAMPLES ON COMMONSENSEQA AND SOCIALIQA

Table 4: Examples of Reasoning Cases from StrategyQA and WinoGrande Datasets. The answer is generated by Gemma2-9B.

| Dataset | StrategyQA | WinoGrande |
|---|---|---|
| **Question** | Is **Ganesha** associated with a Norse god? | It was easy for Angela to become a **vegetarian** although Kayla couldn't do it. _ really missed the taste of chicken. (1) Angela (2) Kayla |
| **Answer** | Ganesha is a **Hindu** god. Norse gods are associated with Norse mythology. Thus, Ganesha is not associated with a Norse god. So the answer is no. | A person is a vegetarian means he does not eat **meat**. A person who really misses the taste of chicken means he likes to eat chicken. Since Angela was able to become a vegetarian but Kayla couldn't do it, Kayla really missed the taste of chicken. So the answer is (2) Kayla. |
| **Answer Type** | Yes / No | Multiple Choice |
| **Answer Token** | no | (2) Kayla |
| **Concept** | Ganesha | vegetarian |
| **Predicted Attr.** | Hindu | meat |
| **General Attr.** | elephant, deity, god | chicken, beef |

Table 5: Examples of Reasoning Cases from CommonsenseQA and SocialIQA Datasets. The answer is generated by Gemma2-9B. In CommonsenseQA and SocialIQA, the entities are often abstract names or professions with no specific meaning. Therefore, we treat the options in the context as attributes, the final predicted option as the predicted attribute, and the remaining options as general attributes.

| Dataset | CommonsenseQA | SocialIQA |
|---|---|---|
| **Question** | The artist was sitting quietly pondering, then suddenly he began to paint when what struck him? (A) sadness (B) anxiety (C) inspiration (D) discomfort (E) insights | remy had a good talk with aubrey so aubrey understood remy better now. How would Remy feel as a result? (1) unsatisfied (2) calm (3) anxious |
| **Answer** | The **artist** was sitting quietly pondering, then suddenly he began to paint when **inspiration** struck him. So the answer is: (C) inspiration. | **Remy** had a good talk with Aubrey. Thus, Aubrey understands Remy better. Remy will feel **calm** as a result. So the answer is: (2) calm. |
| **Answer Type** | Multiple Choice | Multiple Choice |
| **Answer Token** | (C) inspiration | (2) clam |
| **Concept** | artist | Remy |
| **Predicted Attr.** | inspiration | calm |
| **General Attr.** | sadness, anxiety, discomfort | unsatisfied, anxious |

A.5  PATH PATCHING DETAILS

**Counterfactual data generation**  We use GPT-4 to assist in automatically generating the counter-factual data required for path patching, with the prompt shown below. Additionally, we implement a post-processing step: if the predicted token for the counterfactual data matches the prediction for the data under investigation (which would fail to perturb the model's behavior), GPT-4 is prompted to regenerate the counterfactual data.

**Path patching metric**  We use the rate of change in the logits of the predicted token before and after perturbation as the metric for path patching.

---

**Prompt Template for Counterfactual Data Generation**

```
<topic> The particular topic being studied</topic>
<input_sentence> The original sentence provided for
analysis</input_sentence>
<predicted_content> The specific words reflecting model
behavior</predicted_content>
<first_word_predicted> The first word initially predicted by the
model</first_word_predicted>
</Inputs>

<Instructions Structure>
1. Instruct the assistant to begin by analyzing the original input
sentence and why it leads to the specific predicted word.
2. Guide the assistant to think about changes that could alter the
model's prediction.
3. Instruct the assistant to provide the reason for the model's
original prediction.
4. Request the assistant to modify the original sentence so that
the model's prediction changes.
5. Instruct the assistant to explain the modification's rationale,
focusing on why the modified sentence now influences a different
predicted outcome.
6. Ensure the output is formatted in the specified JSON structure.
</Instructions Structure>

<Instructions>
Your task is to analyze and modify a sentence to influence the
predictive behavior of a language model. You will be given a
topic, an input sentence, the specific words predicted by the
model, and the model's first predicted word.

Here is the topic and input sentence to modify:
<topic>{$TOPIC}</topic>
<input_sentence>{$INPUT_SENTENCE}</input_sentence>

Here are the words generated by model given the input sentence:
<predicted_content>{$PREDICTED_CONTENT}</predicted_content>

Here is the first predicted word:
<first_word_predicted>{$FIRST_WORD_PREDICTED}</first_word_predicted>

Follow these steps carefully to complete the task:

1. **Analyze the Original Prediction**: Start by understanding the
**input sentence** and why it leads the model to predict the
**first_word_predicted** as the output under the specific
**topic**. Consider the context, tone, or structure of the sentence
that prompts this specific word choice by the model.
```

---

```
Prompt Template

2. **Plan the Modification**: Think about how you could change the
**input_sentence** minimally (by changing only 3-4 words) to alter
the model's behavior so that it no longer predicts the original
word or instead predicts a word with an opposite meaning.
It's acceptable to change some of the sentence's meaning if it
helps influence the output.

3. **Provide Analysis and Modification**:
   - Write the **reason for the original prediction** based on
   your analysis in Step 1.
   - Rewrite the **input_sentence** in a modified form that
   will change or flip the model's predicted word.

   - Explain your **reason for the modification**, focusing
   on how the changes you made will influence the model to
   predict a different word.

4. **Output the Final Result**: Format your response in JSON,
as shown below:

```json
{
    "Reason for original prediction": "Explain why the original
    input caused the model to predict the initial word.",
    "Modified input": "Write the modified sentence here.",
    "Reason for modification": "Explain why the modified input
    will lead to a different prediction from the model."
}
```

Make sure each section is clear and precise. End your response
with this JSON structure.
</Instructions>
```

Table 6: Example of probing data $X_r$ and counterfactual data $X_c$ generated by GPT-4. Counterfactual data change the model (Gemma2-9B) prediction behavior by applying minimal change to the probing data.

| Data | Model Input | Model Predict |
|------|-------------|---------------|
| $X_r$ | Question: Kendall opened their mouth to speak and what came out shocked everyone. How would you describe Kendall? (1) a very quiet person (2) a very passive person (3) a very aggressive and talkative person Answer: Kendall opened their mouth to speak and what came out shocked everyone. Thus, Kendall is a very __ | aggressive |
| $X_c$ | Question: Kendall opened their mouth to speak and what came out **was softer than expected**. How would you describe Kendall? (1) a very quiet person (2) a very passive person (3) a very aggressive and talkative person Answer: Kendall opened their mouth to speak and what came out **was softer than expected**. Thus, Kendall is a very __ | quiet |

A.6    MORE EXPERIMENTAL RESULT ON GEMMA2-9B

Figure 11 presents the decoding results of the Gemma2-9B model across four commonsense reasoning datasets. The following key observations can be made:

1) The mechanisms of knowledge retrieval and reranking are observed in all datasets, with the decoded outputs from the attention layers containing both predicted attributes $\mathcal{A}_p$ and general attributes $\mathcal{A}_g$.

2) In the StrategyQA and Winogrande datasets, the knowledge augmentation mechanism was identified, as attribute information was decoded from the shallow MLP outputs at the concept token position. However, in SocialIQA (where shallow-layer spikes were observed, though with very low magnitudes) and CommonsenseQA, this mechanism was not evident. We hypothesize that this absence is due to the explicit provision of the required knowledge within the question context, which reduces the model's need to infer additional related knowledge.

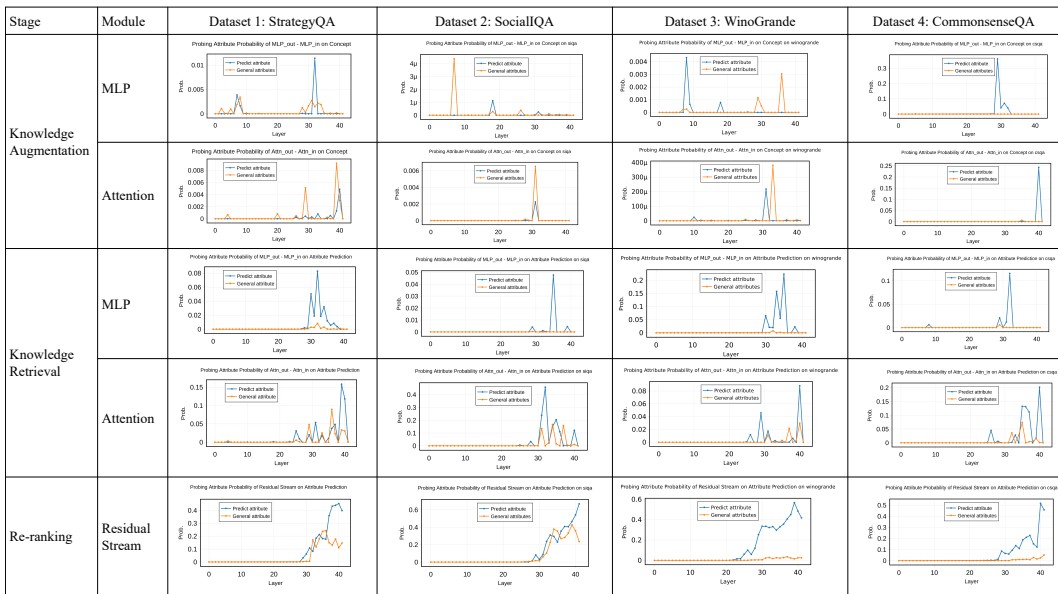

Figure 11: Probing results of Gemma2-9B across four datasets (StrategyQA, WinoGrande, SocialIQA, and CommonsenseQA). The mechanisms of knowledge retrieval and knowledge reranking are observed consistently across all datasets. However, the knowledge augmentation mechanism is absent in SocialIQA and CommonsenseQA (refer to Table 5 for examples), likely because the required knowledge is explicitly provided in the question context, reducing the need for the model to infer additional related knowledge.

Table 7: We fine-tune Llama2-7B/13B on the CommonsenseQA dataset using supervised fine-tuning (SFT) and selectively supervised fine-tuning (SSFT). Here are the capabilities of models on four commonsense reasoning tasks (*e.g.*, Winogrande, CSQA, StrategyQA, and SocialIQA) before and after tuning.

| Models | Tuned Params. | ID Task CSQA | | OOD Task Winogrande | | StrategyQA | | SocialIQA | | Average | |
|---|---|---|---|---|---|---|---|---|---|---|---|
| | | Acc. | Δ | Acc. | Δ | Acc. | Δ | Acc. | Δ | Acc. | Δ |
| Llama2-7B | - | 61.1 | - | 62.5 | - | 53.4 | - | 60.2 | - | 58.7 | - |
| + SFT | 6.7B | 72.3 | +11.2 | 57.8 | -4.7 | 53.5 | +0.1 | 55.7 | -3.0 | 56.2 | -2.5 |
| + SSFT | 0.2B | 73.5 | +12.4 | 63.1 | +0.6 | 56.2 | +2.8 | 63.2 | +3.0 | 61.8 | +3.1 |
| Llama2-13B | - | 68.3 | - | 55.5 | - | 66.0 | - | 67.9 | - | 63.1 | - |
| + SFT | 13B | 78.7 | +10.4 | 55.8 | +0.3 | 64.8 | -1.2 | 64.1 | -3.8 | 61.5 | -1.6 |
| + SSFT | 0.5B | 79.4 | +11.1 | 57.1 | +1.6 | 67.2 | +1.2 | 70.1 | +2.2 | 64.8 | +1.7 |

Table 8: We fine-tune Llama2-7B/13B on the Winogrande dataset using supervised fine-tuning (SFT) and selectively supervised fine-tuning (SSFT). Here are the capabilities of models on four commonsense reasoning tasks (*e.g.*, Winogrande, CSQA, StrategyQA, and SocialIQA) before and after tuning.

| Models | Tuned Params. | ID Task Winogrande | | OOD Task CSQA | | StrategyQA | | SocialIQA | | Average | |
|---|---|---|---|---|---|---|---|---|---|---|---|
| | | Acc. | Δ | Acc. | Δ | Acc. | Δ | Acc. | Δ | Acc. | Δ |
| Llama2-7B | - | 53.4 | - | 61.1 | - | 62.5 | - | 60.2 | - | 61.3 | - |
| + SFT | 6.7B | 74.3 | +20.9 | 56.8 | -4.3 | 64.3 | +1.8 | 59.0 | -1.2 | 60.0 | -1.3 |
| + SSFT | 0.2B | 74.3 | +20.9 | 64.9 | +3.8 | 63.2 | +0.7 | 63.2 | +3.0 | 63.8 | +2.5 |
| Llama2-13B | - | 55.5 | - | 68.3 | - | 66.0 | - | 67.9 | - | 67.4 | - |
| + SFT | 13B | 77.3 | +21.8 | 64.6 | -3.7 | 69.9 | +3.9 | 63.1 | -4.8 | 65.9 | -1.5 |
| + SSFT | 0.5B | 75.6 | +20.1 | 71.7 | +3.4 | 68.6 | +2.6 | 69.2 | +1.3 | 69.8 | +2.4 |

Table 9: We fine-tune Llama2-7B/13B on the SocialIQA dataset using supervised fine-tuning (SFT) and selectively supervised fine-tuning (SSFT). Here are the capabilities of models on four commonsense reasoning tasks (*e.g.*, SocialIQA, CSQA, StrategyQA, and Winogrande) before and after tuning.

| Models | Tuned Params. | ID Task SocialIQA | | OOD Task CSQA | | StrategyQA | | Winogrande | | Average | |
|---|---|---|---|---|---|---|---|---|---|---|---|
| | | Acc. | Δ | Acc. | Δ | Acc. | Δ | Acc. | Δ | Acc. | Δ |
| Llama2-7B | - | 60.2 | - | 61.1 | - | 62.5 | - | 53.4 | - | 59.0 | - |
| + SFT | 6.7B | 74.5 | +14.3 | 65.9 | +4.8 | 61.0 | -1.5 | 52.6 | -0.8 | 59.8 | +0.8 |
| + SSFT | 0.2B | 75.1 | +14.9 | 66.4 | +5.3 | 65.2 | +2.7 | 55.8 | +2.4 | 62.5 | +3.5 |
| Llama2-13B | - | 67.9 | - | 68.3 | - | 66.0 | - | 55.5 | - | 63.3 | - |
| + SFT | 13B | 74.7 | +6.8 | 67.2 | -1.1 | 65.4 | -0.6 | 53.5 | -2.0 | 62.0 | -1.3 |
| + SSFT | 0.5B | 75.1 | +7.2 | 70.7 | +2.4 | 69.9 | +3.9 | 55.8 | +0.3 | 65.5 | +2.2 |

### A.7 MORE EXPERIMENTAL RESULT ON QWEN2.5-72B

We validated three key steps in the internal factual knowledge recall mechanism on Qwen2.5-72B: first, the shallow MLP encodes relevant attribute information (knowledge augmentation); second, the attention heads are responsible for transferring all knowledge to the predicted attribute token position (knowledge retrieval); and finally, the MLP selects the final predicted attribute for output (knowledge reranking).

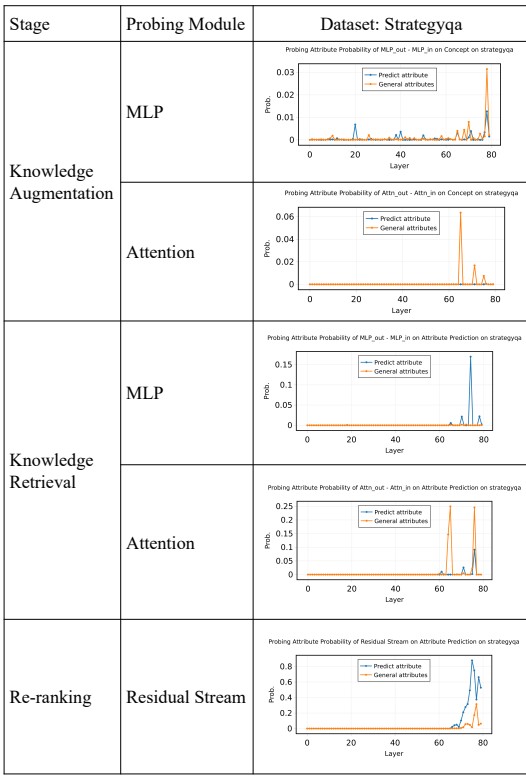

Figure 12: Probing results of Qwen2.5-72B on StrategyQA. The mechanisms of knowledge augmentation, retrieval, and reranking are observed.

Table 10: We fine-tune Qwen2.5-72B on the StrategyQA dataset using supervised fine-tuning (SFT) and selectively supervised fine-tuning (SSFT). Here are the capabilities of models on four commonsense reasoning tasks (e.g., Winogrande, CSQA, StrategyQA, and SocialIQA) before and after tuning.

| Models | Tuned Params. | ID Task | | OOD Task | | | | | | | |
|---|---|---|---|---|---|---|---|---|---|---|---|
| | | StrategyQA | | CSQA | | Winogrande | | SocialIQA | | Average | |
| | | Acc. | Δ | Acc. | Δ | Acc. | Δ | Acc. | Δ | Acc. | Δ |
| Qwen2.5-72B | - | 86.9 | - | 84.1 | - | 78.7 | - | 78.1 | - | 80.3 | - |
| + SFT | 72B | 90.5 | +3.6 | 81.3 | -2.8 | 77.3 | -1.4 | 73.2 | -4.9 | 77.6 | -2.7 |
| + SSFT | 2.5B | 90.0 | +3.1 | 86.6 | +2.5 | 79.0 | +0.3 | 80.0 | +1.9 | 81.9 | +1.6 |

## A.8 PROBING RESULTS ON LLAMA MODELS

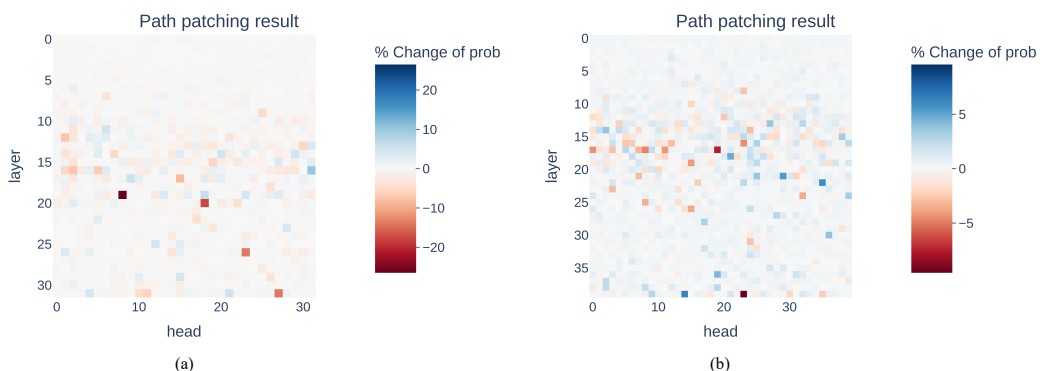

Figure 13: The distribution of the key attention heads for generating attribute in (a) Llama2-7B and (b) Llama2-13B .

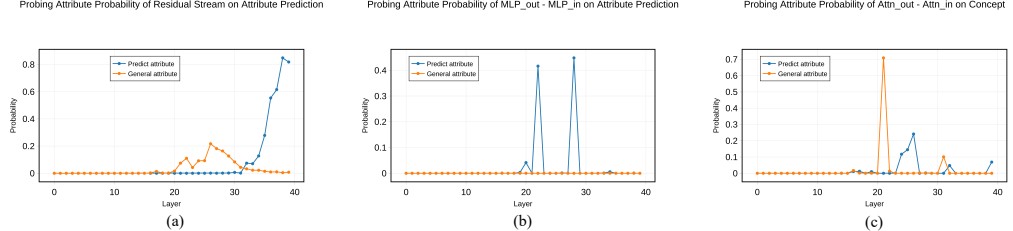

Figure 14: Result from Llama2-13B: (a) Change information of predict attribute and general attribute when predicting the final attribute token. Attribute information contribution from (b) MLP layers and (c) attention layers.

## A.9 PROJECTION OF KEY ATTENTION HEADS FOR CONCLUSION GENERATION

Projection of the key attention heads output when predicting the final conclusion token in Gemma2-9B.

| Head | Top tokens from projecting to vocabulary space |
|------|------------------------------------------------|
| 31.3 (negative) | none, neither, nowhere, nothing, never, no, NONE, neither |
| 26.9 (negative) | neither, contradicts, contradict, unlikely |
| 31.0 (negative) | isn, cannot, wouldn, aren, is, never, doesn, not |
| 31.3 (positive) | naturally, Naturally, future, later, Naturally, obvious, obviously |
| 26.9 (positive) | obviously, umably, presumably, likely, probably, doubtless |
| 31.0 (positive) | would, might, likely, would, probably, Would, expected |

When predicting " Genesha is a Hindu god.", projecting the key attention heads output to vocabulary space, results of Gemma2-9B are shown below:

| Head | Top tokens from projecting to vocabulary space |
|------|------------------------------------------------|
| 25.1 | elephant, Elephant, elefante, religione, Elephants, Hindu, prayers |
| 27.15 | Asian, Asian, Chinese, Asia, Eastern, eastern |
| 29.14 | elephants, elephant, elef, India, Georgia, Maharashtra, Bombay, not |
| 29.15 | Hindu, Indian, India, Hindus, animals, Hinduism, certamen |

## A.10 SAE RELATED DETAILS

This study primarily uses SAE to investigate the information contained in the MLP and residual block outputs at the concept token position. Specifically, we selected the top 64 activated latents (Top-64) based on SAE activations. Since these latents include a substantial number of general-purpose activations (e.g., those representing syntax, specific words, etc.), we employed GPT to automatically analyze whether these activated latents are related to the concept. The prompt used for this analysis is provided below.

---

**Prompt Template**

```
I want to evaluate the relevance of a feature that activates
on certain texts to the concept of `{concept}'.

You will be provided with a possible explanation of the feature,
a set of texts where the feature has been activated, along with
the most activated word(s) in each text.

$<$Example$>$
Concept: `Environmental Protection'

Possible explanation of the feature: feature identifies
texts related to protecting the natural environment.

Activated texts and most activated word:
- must take action to reduce carbon emissions and combat
  climate change. | most activated word: emissions
- Deforestation is a major threat to biodiversity and
  contributes to global warming | most activated word: deforestation
$<$/Example$>$

$<$Expected Output$>$
The feature is highly relevant to the concept of environmental
protection as it identifies texts discussing environmental
issues and solutions.
Relevance Score: 10
$<$/Expected Output$>$

Based on the given explanation of the feature and the activated
texts, please rate the relevance of the feature to the concept
of `{concept}' on a scale of 0 to 10.
- 0: Not at all relevant, the feature is not related to the concept.
- 5: Neutral, the feature is not directly related to the concept
     but share some common traits with the concept, e.g. apple
     and banana are both fruits.
- 10: Very relevant, the feature is directly related to the concept.

Please conclude your response in the following format:
`Relevance Score: [SCORE]', where [SCORE] is an integer between
0 and 10.

Here is the concept: {concept} and the explanation of the feature:

Concept: {concept}

Possible explanation of the feature: {explanation}

Activated texts and most activated word: {texts}
</Instructions>
```

---

### A.11 TRAINING SAE ON LLAMA2-7B

The training code for the Sparse Autoencoder (SAE) is derived from the open-source repository provided by OpenAI (`https://github.com/openai/sparse_autoencoder`), which implements the Top-K activation function to maintain the sparsity of the latent representations. We conducted training of the SAEs using the MLP output obtained from layers 16 and 20 of the LLaMA2-7B model.

The training dataset comprises 2 billion tokens sourced from the Pile dataset, which are organized into sequences of 64 tokens each. Our SAEs are configured to utilize 512,000 latent variables, with the parameter K in the Top-K activation function set to 32. The training parameters include a tensor parallel size of 2, a data parallel size of 8, a batch size of 131,072, and a learning rate of 1.24e-4, which was determined using scaling laws based on the GPT-2 architecture. The SAE was trained for a 1 epoch.

It required approximately 5 hours using 64 A100 GPUs to compute the MLP output for the LLaMA2-7B model across the 2 billion tokens. The training of the SAE itself necessitated around six hours with the utilization of 16 A100 GPUs.

### A.12 MORE CASES AND ANALYSIS OF RECALLED FEATURES ON STRATEGYQA FROM LLAMA2-7B

In this part, more cases of recalled features in Llama-7B are presented in Fig. 15 (a), which corresponds to Layer 20 Fig. 1. We can see that the recalled top features are related to the key concept in input. Furthermore, to compare the precision of recalled features among with/without chain-of-thought, and the proposed SSFT on commonsense reasoning, we collect all the top-4 SAE tokens from the rank$1 - 3$ features of MLP 20 in Llama-7B, and then utilize the GPT-4o to judge if these features are the correct attributes of the input concept. Corresponding precisions are presented in Fig. 15 (b), we can see that with CoT and SSFT, the precision of recalled features are more relevant to the input concept.

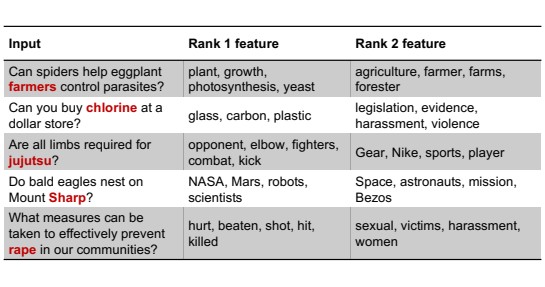

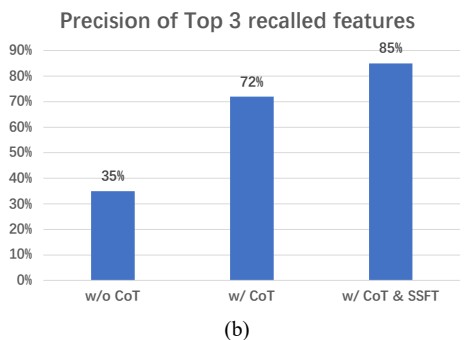

(a)                                                                 (b)

Figure 15: (a) More cases on Llama2-7B that use SAE to explain the MLP information on StrategyQA. (b) Precision of top 3 recalled features under three settings, without CoT prompt, with CoT prompt, and with CoT prompt after SSFT.

| Input | Question: John cannot run the entire length of the track, he had been used to the field. The _ is short.
(1) track (2) field
Answer: A person who cannot run the entire length of a track likely feels uncomfortable or out of practice on a surface that is different from what they are used to. If John had been used to the field, it suggests that he is more accustomed to that environment. Therefore, the track must be |
|---|---|
| SSFT model output | **longer** than the field, making it difficult for him to run its entire length. ✅ |
| Base model output | **shorter** than the field, as he struggles to run its entire length. ❌ |

(a) Case study: output of SSFT and Base model

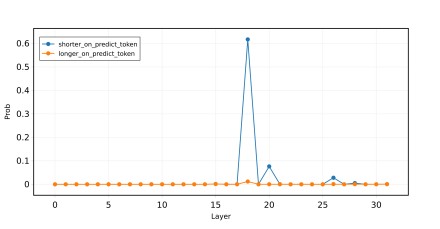

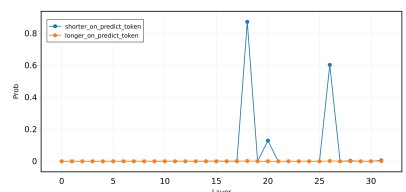

(b) Probing attention layer output for "shorter" and "longer" on SSFT model

(c) Probing attention layer output for "shorter" and "longer" on Base model

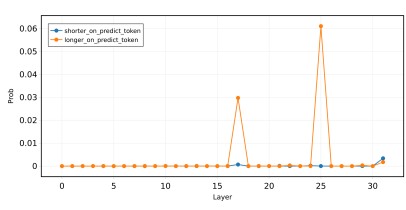

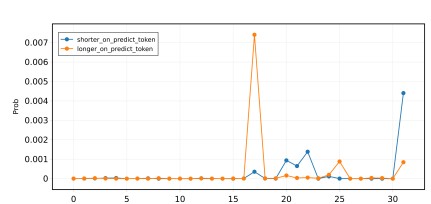

(d) Probing MLP layer output for "shorter" and "longer" on SSFT model

(e) Probing MLP layer output for "shorter" and "longer" on Base model

Figure 16: Comparison between the SSFT and Base models: (a) Case study highlights that the SSFT model correctly predicts the answer, while the Base model fails. (b, c) Probing results for attention layers show enhanced knowledge retrieval in the SSFT model compared to the Base model. (d, e) Probing results for MLP layers demonstrate improved reranking capability in the SSFT model. These findings confirm that the identified modules—attention heads for knowledge retrieval and MLP layers for reranking—are critical for accurate reasoning and were effectively strengthened through SSFT.

