# OpenReview forum: "Deciphering and Enhancing Commonsense Reasoning in LLMs from the Perspective of Intrinsic Factual Knowledge Retrieval"
_ICLR.cc/2025/Conference — Submitted to ICLR 2025_

### Official Review · Reviewer_4kUU · 2024-10-30

**Soundness:** 3
**Presentation:** 2
**Contribution:** 3
**Rating:** 6
**Confidence:** 4

**Summary:**

The paper employs techniques such as Path Patching, Logit Lens, and Sparse AutoEncoders (SAE) to interpret the internal mechanisms of large language models (LLMs) in commonsense reasoning. Specifically, the authors find that: (1) certain positions in the LLM's internal states augment attribute information; (2) later tokens also encode significant attribute information across multiple layers; (3) deeper MLP layers are responsible for retrieving and reranking relevant knowledge; and (4) the final answer is largely a result of transferring and consolidating this accumulated previous information.
Building on these, they analyze the error distribution and discover that most errors stem from the model’s failure to retrieve relevant knowledge. As a result, they selectively fine-tune specific attention heads and MLP layers to enhance the model's ability to retrieve and rerank knowledge. Through experiments they show that the proposed SSFT (Selective Supervised Fine-Tuning) technique outperforms standard SFT while using less than 10% of the model’s parameters.

**Strengths:**

1)	The paper provides a detailed analysis of how the internal states of LLMs function during commonsense reasoning, making the mechanisms of LLMs more transparent. The reviewer believes this will interest a broad audience within the community and serve as a foundation for future research.

2)	The paper introduces a novel fine-tuning technique (SSFT) that selectively fine-tunes the most critical attention heads and MLP layers, achieving improved performance with significantly reduced computational resources.

**Weaknesses:**

1.	The methodology section could benefit from additional details, particularly in Section 3.2, which covers the interpreting module. Providing more explanation on how each of the modules works would be helpful for readers who are not familiar with these techniques, as currently, they may need to consult other papers. The reviewer suggests that adding these details would enhance the coherence and readability of the paper.
2.	The writing in the Experiment Results section could be further refined, especially in sections 4.2, 4.3, and 4.4. There are two issues that the reviewer may think to consider for improvement:
   a.	When introducing specific terms like “concept” and “attributes”, it would be useful to refer back to the Preliminary section or include footnotes to avoid confusion. Readers might be unclear about which specific concepts or attributes you are referring to, given that multiple are mentioned in the preliminary section. Similarly, the distinction between “general attribute” and “predicted attribute”, which appears frequently in these sections, could be clarified to avoid ambiguity.
        b.	Another improvement would be to more explicitly connect each section. Upon first reading, it can be difficult to understand how sections like 4.3 build on 4.2, and I had to spend some time figuring out the relationships. Explaining how each section progresses and relates to the previous one would help guide the reader through the flow of the results.

3.	More explanation could be added to the figures to assist with interpretation. While the figures look visually informative, they take some time to understand. For example, Figure 6a lacks guidance on how it should be interpreted, and adding this clarification would make it easier for readers to grasp the key points.
4.	The logic error example in Section 4.6 is a bit confusing. In the correct answer, it states that Sony sold more units than Sega but concludes that “Sony did not win the war”, which seems contradictory. Given that Sony outsold Sega, shouldn’t the conclusion be “Sony win the war”? This contradiction between the conclusion and the green-underlined statements needs further clarification to resolve the confusion.
5.	There are way too many citation format problems: e.g., line 080, “Wang et al. (2023a)” -> “(Wang et al. 2023a)”. The reviewer hopes that the authors thoroughly check all related issues and fully polish the paper presentation.
6.	In the main experimental results presented in Table 2, the authors only compare the results using the Llama2 as the backbone LLM. Yet they do not analyze the performance across other important relevant LLMs, both open-sourced and proprietary. This omission likely limits the generalizability of the conclusions drawn from the proposed method. The reviewer thus suggests adding more MLLMs for experimental comparison.
7.	More importantly, the reviewer may have identified a problem with the baselines used for comparison by the authors. For instance, they have ignored the performance comparison with all successful RAG-related methods, which should have been included in Table 2. Thus, the reviewer is a little bit skeptical about the consistency of the conclusions. Please provide further detailed explanations.

Overall, the reviewer is open-minded. If the authors can actively and effectively address these concerns, the reviewer would consider raising the rating further.

**Questions:**

1.	Are the probing experiments conducted based on the average results across both datasets, StrategyQA and CommonsenseQA? Have you observed any differences in model behavior between these two datasets?
2.	Could you clarify the distinction between “General Attribute” and “Predicted Attribute”?
3.	What are the exact values for k and l in the SSFT experiment setup? Just to confirm, do these parameters refer to the attention heads and MLP layers you found in section 4.4 as responsible for retrieving and reranking knowledge in the LLM?
4.	Do the authors plan to release the code? the reviewer could not find any provided code or metadata that would allow for a technical review of the details.

---

> ### Author Response · Authors · 2024-11-22
>
> **Part 1/7**
> --
> **Weakness 1**
> > The methodology section could benefit from additional details, particularly in Section 3.2, which covers the interpreting module. Providing more explanation on how each of the modules works would be helpful for readers who are not familiar with these techniques, as currently, they may need to consult other papers. The reviewer suggests that adding these details would enhance the coherence and readability of the paper.
>
> ***Ans to Weakness 1:***
>
> To address the concerns, we will add detailed explanations of the utility of each module in the **Methods** section and provide illustrative examples to make the concepts more accessible. Below is the corresponding table:
>
> | **Interpreting Module**       | **Input**                                             | **Output**                                                                                                 | **Conclusion (Example)**                                                                                                                                         |
> |-------------------------------|------------------------------------------------------|-----------------------------------------------------------------------------------------------------------|------------------------------------------------------------------------------------------------------------------------------------------------------------------|
> | **Path Patching**             | Commonsense reasoning data + counterfactual data    | Distribution of head importance                                                                           | For example, with "Ganesha is a Hindu god," at the "a" token position, path patching reveals which attention heads in the LLM are critical for predicting "Hindu." |
> | **Logit Lens**                | Commonsense reasoning data + probing attributes     | Attribution of investigated attribute token within different modules (MLP, attention, residual block)              | For example, with "Ganesha is a Hindu god," at the "a" token position, this method shows which attention layers transport related attributes (knowledge retrieval), and which MLP layers perform re-ranking to generate the predicted attribute. |
> | **SAE (Sparse Autoencoder)**  | Commonsense reasoning data + specific layer ID      | Identifies what information is encoded in the output of a specific MLP layer                              | For example, with "Ganesha is a Hindu god," at the "Ganesha" token position, SAE helps decompose the MLP output to determine what attribute-related information is encoded. |
> | **Attention Head Pattern Analysis** | Commonsense reasoning data + specific head ID         | Attention score and projection of head outputs into vocabulary space                                     | For the example "Ganesha is a Hindu god," using the "attention head pattern analysis module" at the "is" token position, we analyzed the heads identified by path patching. This allows us to determine which important heads transported information, from which tokens, and what specific information was transported. |
>
> These additions aim to clarify the functionality of each module and make the results more interpretable. They will be updated in the revised version of the paper, which will be submitted soon.

---

> ### Author Response · Authors · 2024-11-22
>
> **Part 2/7**
> --
> **Weakness 2. a**
> > When introducing specific terms like “concept” and “attributes”, it would be useful to refer back to the Preliminary section or include footnotes to avoid confusion. Readers might be unclear about which specific concepts or attributes you are referring to, given that multiple are mentioned in the preliminary section. Similarly, the distinction between “general attribute” and “predicted attribute”, which appears frequently in these sections, could be clarified to avoid ambiguity.
>
> ***Ans to Weakness 2. a:***
>
> Thank you for your valuable suggestion. We have revised the paper to address the concerns as the following:
>
> **1. Enhanced Definitions in the Preliminary Section**
>
> We have introduced the notation $\mathcal{C}$ to explicitly represent the previously defined “concept,” which corresponds to Concept A and Concept B in the data template. Similarly, we now use $\mathcal{A}$ to represent “attributes,” corresponding to Attribute A and Attribute B.
>
> **2. Definitions of Predictive Attribute and General Attribute**
>
> We have formally defined **predictive attribute** and **general attribute** in the Preliminary section. To aid understanding, we have also included a visual illustration available at the following anonymous link: [Illustration for Attributes](https://anonymous.4open.science/r/ecf9458a-50f9-4052-bdb8-0e79db28ed17/illustration4attributes.pdf) (if the image fails to load, refreshing the page or downloading it locally should resolve the issue.)
>
> - **Predicted Attribute**: Denoted as $\mathcal{A}_p$, this represents the attribute with the highest probability output by the model in a specific commonsense reasoning context.
> - **General Attribute**: Denoted as $\mathcal{A}_g$, this includes other related attributes that the model might consider in the same context but does not select as the primary prediction.
>
> For example, here is a question and model reasoning output from StrategyQA:
> - **Question**: “Q: Is Ganesha associated with a Norse god?”
> - **Model Answer**: “Ganesha is a Hindu god.”
>
> Here, for the concept “Ganesha,” the predicted attribute $\mathcal{A}_p$ is **Hindu**, while the general attributes $\mathcal{A}_g$ include related attributes such as **god**, **deity**, and **elephant**.
>
> **3. Updates to Experimental Descriptions**
> We have revised all references to “concept” and “attribute” in the experiment sections to align with the definitions provided in the Preliminary section. For instance, the following excerpt demonstrates how the revised descriptions are structured:
>
> **Revised Version Excerpt**:
> To investigate the origin of $\mathcal{A}$, we analyzed the probabilities of both $\mathcal{A}_g$ and $\mathcal{A}_p$ attributes across different layers. Our objective was to determine whether the model encodes information about $\mathcal{A}$ that is relevant to $\mathcal{C}$. Figure 3(a) depicts the amount of $\mathcal{A}$ information encoded at the $\mathcal{C}$ token position across layers. For comparison, we also evaluated the quantity of $\mathcal{A}$ information at the token position preceding $\mathcal{C}$. The findings reveal that the $\mathcal{C}$ token position encodes a significant amount of $\mathcal{A}$ information, with notable concentrations in the model’s shallow layers (8-12) and deeper layers (33-39).
>
>
> We will soon submit the revised version of the paper incorporating these updates. Thank you again for your feedback.

---

> ### Author Response · Authors · 2024-11-22
>
> **Part 3/7**
> --
> **Weakness 2. b**
>
> > Another improvement would be to more explicitly connect each section. Upon first reading, it can be difficult to understand how sections like 4.3 build on 4.2, and I had to spend some time figuring out the relationships. Explaining how each section progresses and relates to the previous one would help guide the reader through the flow of the results.
>
> ***Ans to Weakness 2. b:***
>
> Thank you for your feedback. We follow the philosophy of the model's inner reasoning process to organize the subsections in the Experiment section. Below is the content we include to guide readers through the flow of the results:
>
> **Section Interconnections**
>
> 1. **Knowledge Augmentation (Section 4.2)**:
>    We begin by investigating how the model leverages the input context to identify attributes related to key concepts. This analysis lays the foundation for subsequent exploration of retrieval and reasoning tasks.
>
> 2. **Knowledge Augmentation Across Layers (Section 4.3)**:
>    Building on the findings from Section 4.2, we examine how attribute information evolves across different layers of the model as it processes the full question context. This section demonstrates how the initial set of attributes is refined layer by layer to create an enhanced knowledge set for prediction.
>
> 3. **Knowledge Retrieval (Section 4.4)**:
>    After analyzing the augmentation phase, we delve into how the model employs multi-layer mechanisms to re-rank and retrieve the most relevant attributes from the augmented knowledge. This section highlights the retrieval process that bridges knowledge augmentation and answer generation.
>
> 4. **Answer Generation (Section 4.5)**:
>    With the necessary knowledge retrieved, we analyze how the model uses this information to generate final answers. This section focuses on the specific roles of key layers and attention heads in synthesizing the retrieved attributes into accurate outputs.
>
> 5. **Selective Supervised Fine-Tuning (Section 4.6)**:
>    Finally, informed by the insights gained in the preceding sections, we propose and validate a **Selective Supervised Fine-Tuning (SSFT)** strategy. This strategy leverages our understanding of the model’s internal mechanisms to enhance reasoning performance, both within the training domain and on out-of-domain tasks.
> ---
> **Weakness 3**
> > More explanation could be added to the figures to assist with interpretation. While the figures look visually informative, they take some time to understand. For example, Figure 6a lacks guidance on how it should be interpreted, and adding this clarification would make it easier for readers to grasp the key points.
>
> ***Ans to Weakness 3:***
>
> Thank you for your feedback. We will revise the figure captions in the updated version of the paper to include more detailed explanations. Below, we provide examples of two revised captions:
>
> **Figure 1: Deciphered Commonsense Reasoning Process in LLMs**
> **Original Caption**:
> The five stages of the process are depicted through the example of addressing a reasoning question, as presented in the leftmost column, while the corresponding generated answer is showcased in the rightmost column, utilizing the Llama2-7B model.
> **Revised Caption**:
> This figure illustrates the five stages of the commonsense reasoning process in LLMs using an example question. The leftmost column presents the input question, while the rightmost column displays the final answer generated by the Llama2-7B model. The central columns depict the processing sequence across model layers, highlighting critical steps such as knowledge augmentation, retrieval, and re-ranking. Each stage is explicitly labeled and mapped to specific model layers, providing a clear explanation of how the reasoning process aligns with the model's architecture.
>
> **Figure 6a: Key Attention Heads for Generating the Conclusion Token**
> **Original Caption**:
> "Key attention heads for generating the conclusion token."
> **Revised Caption**:
> This figure illustrates the key attention heads involved in generating the conclusion token. Attention heads from intermediate to later layers (25-30) are shown to contribute significantly by transferring accumulated semantic information from prior layers to produce the final conclusion. The size of each node represents the strength of its contribution, visually emphasizing the importance of specific attention heads in this process.

---

> ### Author Response · Authors · 2024-11-22
>
> **Part 4/7**
> --
> **Weakness 4**
> > The logic error example in Section 4.6 is a bit confusing. In the correct answer, it states that Sony sold more units than Sega but concludes that “Sony did not win the war”, which seems contradictory. Given that Sony outsold Sega, shouldn’t the conclusion be “Sony win the war”? This contradiction between the conclusion and the green-underlined statements needs further clarification to resolve the confusion.
>
> ***Ans to Weakness 4:***
> Thanks for the suggestion. The contradiction in the logic error example has been corrected, and the updated results are now available at the following anonymous link: [Error Case Distribution](https://anonymous.4open.science/r/ecf9458a-50f9-4052-bdb8-0e79db28ed17/error_case_distribution.pdf).
>
> ---
>
> **Weakness 5**
> > There are way too many citation format problems: e.g., line 080, “Wang et al. (2023a)” -> “(Wang et al. 2023a)”. The reviewer hopes that the authors thoroughly check all related issues and fully polish the paper presentation.
>
> ***Ans to Weakness 5:***
>
> We will thoroughly review and correct all citation-related problems in the revised version to ensure consistency and improve the overall presentation of the paper.

---

> ### Author Response · Authors · 2024-11-22
>
> **Part 5/7**
> --
> **Weakness 6**
> > In the main experimental results presented in Table 2, the authors only compare the results using the Llama2 as the backbone LLM. Yet they do not analyze the performance across other important relevant LLMs, both open-sourced and proprietary. This omission likely limits the generalizability of the conclusions drawn from the proposed method. The reviewer thus suggests adding more MLLMs for experimental comparison.
>
> ***Ans to Weakness 6:***
>
> Below, we provide our response and additional experimental results.
>
> **1. Multi-modal Models and Broader Exploration**
>
> We acknowledge that our current work does not extensively cover multi-modal LLMs. As this is an area where we currently have limited expertise, we plan to explore and validate our approach with multi-modal models in our future work.
>
> **2. Additional Experiments for Generalization**
>
> To address concerns about generalization, we conducted additional experiments using the **Qwen-2.5-72B** model. The results of selective supervised fine-tuning (SSFT) and probing on the **StrategyQA dataset** are shown below:
>
> | **Models**       | **StrategyQA** | **CSQA (OOD)** | **WinoGrande (OOD)** | **SocialIQA (OOD)** | **Avg on OOD Task** |
> |-------------------|----------------|----------------|-----------------------|---------------------|---------------------|
> | **Qwen-2.5-72B**  | 86.9           | 84.1           | 78.7                 | 78.1                | 80.3                |
> | + SFT            | 90.5 (+3.6)    | 81.3 (-2.8)    | 77.3 (-1.4)          | 73.2 (-4.9)         | 77.6 (-2.7)         |
> | + SSFT           | 90.0 (+3.1)    | 86.6 (+2.5)    | 79.0 (+0.3)          | 80.0 (+1.9)         | 81.9 (+1.6)         |
>
> The probing results on **Qwen-2.5-72B** can be found here: [Probing Results for Qwen-2.5-72B](https://anonymous.4open.science/r/ecf9458a-50f9-4052-bdb8-0e79db28ed17/qwen72b_probing.pdf).
>
> Additionally, we performed **probing and SSFT experiments** on two new datasets: **SocialIQA** and **WinoGrande**. The results can be found at the following links:
> - **Probing Results**: [Probing Results for Three Datasets](https://anonymous.4open.science/r/ecf9458a-50f9-4052-bdb8-0e79db28ed17/probing_3_dataset_on_gemma2-9b.pdf) (Consistent curve trends, e.g., attribute knowledge acquisition,  as reported in the paper.)
> - **SSFT Results**: [SSFT Results on SIQA and WinoGrande](https://anonymous.4open.science/r/ecf9458a-50f9-4052-bdb8-0e79db28ed17/ssft_on_siqa_winogrande_llama.pdf) (SSFT performs better than SFT on out-of-domain datasets, while achieving comparable performance on in-domain datasets.)
>
>
> These results demonstrate the generalizability of our method:
> - **Across Model Sizes and Families**: We verified the validity of our method on models ranging from 7B to 72B across different model families.
> - **Across Datasets**: Both interpretability (probing) results and SSFT improvements were consistently observed on multiple commonsense reasoning datasets.
>
> **3. Limitations of Closed-Source Models**
>
> Regarding the application of our method to proprietary models, we acknowledge that our interpretability-related experiments are currently limited to open-sourced models. This is because analyzing the internal behavior of model components and improving performance by fine-tuning parameters **require access to the model’s architecture and weights**, which is not feasible with proprietary models.
>
> However, we have considered the possibility of applying our method to proprietary models. One of the key insights from our work is that reasoning errors in LLMs often stem from the **re-ranking mechanism**, where the model retrieves unrelated knowledge instead of the correct one. Based on this finding, we propose that it might be possible to analyze commonsense reasoning failures in proprietary models by investigating their outputs for specific cases. For example, using prompts, we can examine whether a model is aware of the correct knowledge (e.g., by presenting a factual statement like "Persephone is the Greek goddess of the underworld" and asking the model to verify its correctness). If the correct knowledge is known but not applied in reasoning, the issue is likely attributable to the re-ranking mechanism.
>
> In future work, we plan to extend our analysis to proprietary models, investigating the causes of reasoning errors and exploring how prompt-based approaches might enhance their commonsense reasoning capabilities.

---

> ### Author Response · Authors · 2024-11-22
>
> **Part 6/7**
> --
> **Weakness 7**
>
> > More importantly, the reviewer may have identified a problem with the baselines used for comparison by the authors. For instance, they have ignored the performance comparison with all successful RAG-related methods, which should have been included in Table 2. Thus, the reviewer is a little bit skeptical about the consistency of the conclusions. Please provide further detailed explanations.
>
> ***Ans to Weakness 7:***
>
> We apologize for the misunderstanding. Actually, we use SSFT to improve the model's internal knowledge recall capabilities, particularly focusing on the **re-ranking process**. **RAG** enhances model performance by incorporating external knowledge.
>
> To compare our approach with RAG-related methods, we first utilized GPT-4 to extract the factual knowledge relevant to the entities in each question from the StrategyQA dataset. This factual knowledge is represented as simple single sentences, similar to triples, and does not include explanations. Some factual knowledge is directly relevant to answering the questions (e.g., "Ganesha is a Hindu god"), while others are irrelevant or misleading (e.g., "Ganesha looks like an elephant").
>
> Subsequently, using FaissVectorStore provided by **LlamaIndex** and the UAE model, we generated embeddings for these factual knowledge statements to construct a knowledge base. During inference, relevant factual knowledge is retrieved and added to the context to supplement the model’s predictions. We applied basic vector retrieval, restricting the number of retrieved factual knowledge items to three. Factual knowledge with similarity scores below 0.6 was filtered out to avoid irrelevant information. Using this approach, we compared the following configurations:
>
> - **Llama2-7B + RAG**: 74.7
> - **Llama2-7B + SSFT**: 78.5
> - **Llama2-7B + SSFT + RAG**: 80.4
>
> During testing, the retrieval recall rate of RAG was 94%. However, for Llama2-7B, the factual knowledge provided by RAG only led to limited performance improvements, even when combined with SSFT. This is primarily because answering commonsense reasoning questions requires not only correctly retrieving knowledge but also accurately **utilizing** it.
>
> For example:
> **Q**: Yes or no: Would Persephone be a good consultant to a landscape architect?
> The retrieved knowledge includes:
> - "Persephone is the god of spring."
> - "Persephone is the Greek goddess."
> - "Persephone is the god of the underworld."
>
> Despite this, the model’s response remained **no**. However, after applying SSFT, the model was able to answer **yes** correctly. This indicates that the bottleneck is not in knowledge retrieval but in the model's ability to **effectively use the retrieved knowledge**. This aligns with our findings in the paper, where the primary issue in failure cases was observed to occur during the **re-ranking stage**.
>
> Since SSFT not only enhances the model’s knowledge recall capabilities but, more importantly, improves its re-ranking abilities, it provides a much more substantial boost to the model's overall performance.

---

> ### Author Response · Authors · 2024-11-22
>
> **Part 7/7**
> --
> **Question 1**
> > Are the probing experiments conducted based on the average results across both datasets, StrategyQA and CommonsenseQA? Have you observed any differences in model behavior between these two datasets?
>
> ***Ans to Question 1:***
>
> The probing experiments results presented in the paper were based on the StrategyQA dataset. We have now supplemented the paper with additional results from CommonsenseQA for comparison.
>
> Unlike StrategyQA, in CommonsenseQA **reasoning is largely based on knowledge explicitly provided in the context**, as the example shown below:
>
> - **Question**: The artist was sitting quietly pondering, then suddenly he began to paint when what struck him?
>   (A) sadness (B) anxiety (C) inspiration (D) discomfort (E) insights
> - **Model's Answer**: The artist was sitting quietly pondering, then suddenly he began to paint when **inspiration** struck him. So the answer is: (C) inspiration.
>
> In this case, we probed the "**inspiration**" token, akin to the "attribute" defined in the paper, to investigate how the model leveraged contextual information to infer the most appropriate attribute.
>
> The probing results for CommonsenseQA can be found at the following anonymous link: [Link to Results](https://anonymous.4open.science/r/ecf9458a-50f9-4052-bdb8-0e79db28ed17/probing_strate_csqa_on_gemma2-9b.pdf). Based on these results, we summarize the model’s internal reasoning process for CommonsenseQA:
>
> - **Knowledge augmentation**: At the position of the predicted token, attention heads in the intermediate-to-later layers directly focus on the option tokens, transferring their associated knowledge (e.g., "sadness," "anxiety," "inspiration," and "discomfort") to the residual stream at the predicted token position.
> - **Re-ranking**: Subsequently, the intermediate-to-later MLP layers perform re-ranking, prioritizing and selecting the most contextually appropriate attribute. Ultimately, the model predicts "inspiration" as the final output.
>
> We observed that the knowledge retrieval and re-ranking stages align with the findings reported in the paper. However, the **knowledge augmentation phenomenon is less pronounced** for CommonsenseQA. We hypothesize this is because the required knowledge is explicitly provided in the question context, reducing the model's need to infer additional related knowledge.
>
> ---
>
> **Question 2**
> > Could you clarify the distinction between “General Attribute” and “Predicted Attribute”?
>
> ***Ans to Question 2:***
>
> - The token with the **highest probability predicted by the model** is referred to as the **predicted attribute**.
> - Other relevant attributes that the model could potentially predict (but did not) are referred to as **general attributes**.
>
> **Visual Aid for Attribute Distinction**
> To further clarify these distinctions, we have prepared a visual illustration. The image provides an example-based explanation of predicted attributes versus general attributes, as described above. The image is available at this anonymous link: [Illustration of Attributes](https://anonymous.4open.science/r/ecf9458a-50f9-4052-bdb8-0e79db28ed17/illustration4attributes.pdf) (if the image fails to load, refreshing the page or downloading it locally should resolve the issue).
>
> ---
>
> **Question 3**
>
> > What are the exact values for k and l in the SSFT experiment setup? Just to confirm, do these parameters refer to the attention heads and MLP layers you found in section 4.4 as responsible for retrieving and reranking knowledge in the LLM?
>
> ***Ans to Question 3:***
>
> We apologize for any confusion caused. For the SSFT experiment setup, the parameter configurations are as follows:
>
> - **Llama-2-7B**: The number of fine-tuned attention heads \( k = 32 \), and the number of fine-tuned MLP layers \( l = 2 \).
> - **Llama-2-13B**: The configuration is \( k = 64 \), and \( l = 2 \).
>
> Yes, the attention heads we fine-tuned are responsible for **retrieving knowledge**, while the MLP layers are responsible for **re-ranking** knowledge, as identified in Section 4.4 of the paper.
>
> ---
>
> **Question 4**
> > Do the authors plan to release the code? the reviewer could not find any provided code or metadata that would allow for a technical review of the details.
>
> ***Ans to Question 4:***
>
> Yes, we plan to release the code to ensure transparency and facilitate further research. Additionally, we will include more detailed prompts used for GPT-4-assisted automatic interpretability in the appendix to enhance the reproducibility and technical validity of our work.

---

### Official Review · Reviewer_DWRJ · 2024-11-04

**Soundness:** 3
**Presentation:** 2
**Contribution:** 2
**Rating:** 3
**Confidence:** 3

**Summary:**

The paper investigates the commonsense reasoning processes within large language models (LLMs) by analyzing hidden states across layers and token positions using interpretability techniques such as Path Patching, Logit Lens, and Sparse Autoencoder. The authors propose a five-stage reasoning model within LLMs and identify specific failures in knowledge retrieval. To enhance reasoning performance, they introduce a selective supervised fine-tuning (SSFT) approach that targets specific layers of the model. Experiments demonstrate slight improvements over full-model fine-tuning, particularly in out-of-domain tasks, suggesting potential benefits of selective tuning for commonsense reasoning tasks.

**Strengths:**

•	Application of Interpretability Techniques: The use of Path Patching, Logit Lens, and Sparse Autoencoder offers a diverse set of tools to interpret commonsense reasoning within LLMs.

•	Focused Analysis on Model Components: The identification of key components, such as attention heads and MLP layers, that influence commonsense reasoning could inform future efforts to improve model architectures.

•	Resource-Efficient Fine-Tuning: Introducing SSFT presents a potentially more efficient alternative to full-model fine-tuning, showing moderate improvements in certain scenarios.

**Weaknesses:**

•	Limited Novelty: The proposed reasoning process and interpretability insights do not substantially differ from existing concepts, limiting the originality of the work.

•	Marginal Improvements from SSFT: The performance gains from selective fine-tuning are minimal, making it unclear whether the approach offers significant advantages over existing methods.

•	Surface-Level Interpretability Analysis: The interpretability techniques are applied without providing deep insights into the specific reasons behind the model’s failures in commonsense reasoning tasks.

•	Narrow Experimental Scope: The experiments focus on a limited set of datasets, lacking diversity that could provide a more comprehensive evaluation of the model’s reasoning abilities across different contexts

**Questions:**

1. Have the authors tested the SSFT approach on larger models or more complex tasks to assess its broader applicability?

2. How does this work differentiate itself from prior studies on retrieval-augmented generation and knowledge editing, and what unique insights does it provide on commonsense reasoning?

3. Are there plans to extend the interpretability analysis to explore deeper relationships between reasoning components, such as hierarchical or causal connections?

---

> ### Author Response · Authors · 2024-11-22
>
> **Part 1/6**
> --
> **Weakness 1**
> > Limited Novelty: The proposed reasoning process and interpretability insights do not substantially differ from existing concepts, limiting the originality of the work.
>
> ***Ans to Weakness 1-part1:***
>
> We compared our study with existing studies on the interpretability of factual knowledge across three dimensions: **Reasoning Process**, **Interpreting Tools**, and **Key Findings**. The differences are summarized in the table below:
>
> | **Paper Title** | **Reasoning Process**                                            | **Interpreting Tools**                                  | **Key Findings**                                                                                     |
> |-------------------------------------------------------------------|-------------------------------------------|-----------------------------------------------------------------------------------------------------|--------------------------------------|
> | **Dissecting Recall of Factual Associations in Auto-Regressive Language Models** (EMNLP 2023) | Knowledge augmentation $\rightarrow$ knowledge retrieval | Logit lens, knockout                                                                               | LLMs store factual knowledge in mid-layer MLPs, then retrieve them by attention layers.               |
> | **Locating and Editing Factual Associations in GPT** (NIPS 2022) | Knowledge augmentation $\rightarrow$ knowledge retrieval | Causal Tracing                                                                                     | LLMs store factual knowledge in mid-layer MLPs, then retrieve them by attention layers.               |
> | **LLM Factoscope: Uncovering LLMs’ Factual Discernment through Measuring Inner States** (ACL 2024) | None                                        | Logit lens, neuron activation pattern                                                              | LLMs have distinguishable activation patterns between factual and non-factual content.                |
> | **Attention Satisfies: A Constraint-Satisfaction Lens on Factual Errors of Language Models** (ICLR 2024) | Knowledge retrieval                        | Attention pattern analysis                                                                         | LLMs should attend to entities to retrieve correct factual knowledge.                                 |
> | **Neuron-Level Knowledge Attribution in Large Language Models** (EMNLP 2024) | Knowledge augmentation                     | Logit lens                                                                                        | LLMs store knowledge both in attention heads and MLPs.                                                |
> | **Ours**                                                        | Knowledge augmentation $\rightarrow$ knowledge retrieval $\rightarrow$ knowledge re-ranking | Logit lens, path patching, knockout, SAE, attention pattern analysis                              | LLMs store factual knowledge in shallow-layer MLPs, then retrieve them by attention layers, finally re-rank them by deep-layer MLPs. |
>
>
> We summarize the **Interpretability Insights** as follows:
>
> 1. **Reasoning Process**: We identify a novel **re-ranking process**, which has not been explicitly addressed in prior studies.
> 2. **Interpreting Tools**: With the aim to explore the *token-by-token* reasoning process on *realistically queried* sentences, we developed and utilized customized interpretability tools such as path patching and SAE. In contrast, previous studies primarily focused on the *single-token* reasoning process using *triplet* templates and tools like the logit lens.
> 3. **Key Findings**: Our results reveal that for LLMs to excel in commonsense reasoning, they must not only **acquire knowledge** but also **learn to effectively utilize it** through re-ranking.

---

> ### Author Response · Authors · 2024-11-22
>
> **Part 2/6**
> --
> ***Ans to Weakness 1-part2:***
>
> For example, consider the following case in Figure 7:
>
> | **Question**                                                    | **Model Output (Gemma-2-9B)**                                                                                          |
> |------------------------------------------------------------------|------------------------------------------------------------------------------------------------------------------------|
> | Q: Yes or no: Would Persephone be a good consultant to a landscape architect? | Persephone is the Greek goddess of the **underworld**. Landscape architects design gardens. Thus, Persephone would not be a good consultant to a landscape architect. So the answer is no. |
>
> In this case, the model provides an incorrect answer. During the reasoning process, it recalls the knowledge "Persephone is the Greek goddess of the **underworld**," which is factually correct but not the most relevant attribute for this context. Instead, the model should have recalled "Persephone is the Greek goddess of the **spring**."
>
> Through probing and prompting, we confirmed that the model knows the fact "Persephone is the Greek goddess of the **spring**," but fails to utilize this knowledge in the reasoning process. This demonstrates that in commonsense reasoning, models must go beyond simply acquiring knowledge—they must also learn how to effectively apply it.
>
> ---
> **Weakness 2**
> > Marginal Improvements from SSFT: The performance gains from selective fine-tuning are minimal, making it unclear whether the approach offers significant advantages over existing methods.
>
> ***Ans to Weakness 2:***
>
> We apologize for any misunderstanding caused by the initial presentation of the results in the paper. In fact, SSFT achieved significant improvements over SFT on two out-of-domain (OOD) tasks (+9.3% / +8.4%) while delivering comparable performance on the in-domain (ID) task (+1.2%). These results validate the advantages of SSFT, which leverages the interpreted mechanism to enhance the inherent commonsense reasoning abilities of LLMs, thereby improving their generalization to unseen domains (OOD).
>
> We have updated the organization of the tables and figures in the paper. The revised results can be found at the following anonymous link: [Updated Table](https://anonymous.4open.science/r/ecf9458a-50f9-4052-bdb8-0e79db28ed17/ssft_on_strategyqa_csqa_llama_7b.pdf). (Results are provided via anonymous links as OpenReview does not support direct image uploads. If the image in the link does not load, refreshing or downloading the link could resolve the issue.)
>
> We have identified that commonsense reasoning in LLMs involves three core mechanisms: **knowledge augmentation**, **knowledge retrieval**, and **knowledge re-ranking**, which are consistent across various tasks. Enhancing these mechanisms using a single dataset (e.g., StrategyQA) can effectively improve the model's commonsense reasoning capabilities on other datasets. Unlike SFT, which adjusts all parameters indiscriminately, mechanism-driven interpretability followed by SSFT enables precise and targeted adjustments to these critical mechanisms within LLMs.

---

> ### Author Response · Authors · 2024-11-22
>
> **Part 3/6**
> --
> **Weakness 3**
>
> > Surface-Level Interpretability Analysis: The interpretability techniques are applied without providing deep insights into the specific reasons behind the model’s failures in commonsense reasoning tasks.
>
> ***Ans to Weakness 3:***
>
> We have conducted an analysis of the specific reasons behind the model's failures in commonsense reasoning. The results are presented in Figure 7. For your convenience, we provide an anonymous link: [Error Case Distribution](https://anonymous.4open.science/r/ecf9458a-50f9-4052-bdb8-0e79db28ed17/error_case_distribution.pdf) (if the image does not load, refreshing or downloading the link could resolve the issue).
>
> Our statistical analysis indicates that the majority of errors stem from the use of irrelevant or incorrect factual knowledge as a reference during the reasoning process. Using the same case, the model utilized "Persephone is the Greek goddess of the **underworld**" to generate an incorrect reasoning conclusion. While this knowledge is factually accurate, the attribute that should have been recalled is "Persephone is the Greek goddess of the **spring**."
>
> **Interpretability Insights into Model Failures**
> The mechanism of commonsense reasoning contains three stages:
> 1. **Knowledge Augmentation**: Shallow MLP layers encode relevant knowledge into the residual stream on concept token position.
> 2. **Knowledge Retrieval**: The model’s attention heads retrieve multiple relevant attributes and transport them to the current prediction position.
> 3. **Re-ranking Mechanism**: The MLP layers re-rank these attributes to select the most contextually appropriate attribute for the prediction.
>
> The experiments reveal that the majority of errors occur during the re-ranking stage. The attention layers in the model successfully transport relevant attributes (e.g., "underworld" and "spring") to the predicted token position. However, the MLP layer responsible for re-ranking selects incorrect "underworld" for output. This provides the insight that while LLMs have already acquired the relevant knowledge, they often fail to effectively utilize it.
>
> **Case Study**:
>
> A specific example demonstrates this issue:
>
> | **Question** | **Model output (Gemma-2-9B)** |
> |--------------|-------------------------------|
> | Q: Yes or no: Would Persephone be a good consultant to a landscape architect? | Persephone is the Greek goddess of the **underworld**. Landscape architects design gardens. Thus, Persephone would not be a good consultant to a landscape architect. So the answer is no. |
>
> Our experiments reveal that at the concept token position (i.e., "Persephone"), the MLP in layer 9 has already encoded information about "spring" into the residual stream. Moreover, at the predicted token position, the attention layer in layer 36 enhances the "spring" information. However, the MLP in layer 34, which plays a dominant role in the re-ranking mechanism, ultimately causes the model to predict an irrelevant attribute. These findings indicate that while the model has already **acquired** the relevant knowledge, it fails to effectively **utilize** it. Additionally, our approach can accurately identify the cause of such errors.
>
> - The probing curves for this case, available at this anonymous link ([Probing Results for Error Case](https://anonymous.4open.science/r/ecf9458a-50f9-4052-bdb8-0e79db28ed17/spring_god.pdf)), illustrate the contribution of the MLP layer to the error.

---

> ### Author Response · Authors · 2024-11-22
>
> **Part 4/6**
> --
> **Weakness 4**
>
> > Narrow Experimental Scope: The experiments focus on a limited set of datasets, lacking diversity that could provide a more comprehensive evaluation of the model’s reasoning abilities across different contexts
>
> ***Ans to Weakness 4-part1:***
>
> We conducted additional experiments on the **WinoGrande** and **SocialIQA** datasets. These datasets provide complementary perspectives as they differ significantly in format from the dataset used in our paper. We provide a detailed overview and features of these datasets.
>
> **WinoGrande**:
> The factual knowledge required by WinoGrande is relatively straightforward compared to the Wikipedia-based knowledge in StrategyQA. For example:
> |  Question | Model output (Gemma-2-9B) |
> | - | - |
> | It was easy for Angela to become a vegetarian although Kayla couldn't do it. _ really missed the taste of chicken. (1) Angela (2) Kayla  |  A person is a vegetarian means they do not eat **meat**. A person who really misses the taste of chicken likely enjoys eating chicken. Since Angela became a vegetarian and Kayla could not, Kayla really missed the taste of chicken. |
>
> In this example, we probed the token "**meat**", akin to the "attribute" defined in the paper, to investigate how the model encodes factual knowledge such as "a vegetarian does not eat meat."
>
> **SocialIQA**:
> In SocialIQA, the reasoning is based on knowledge provided in the context, whereas our original experiments relied on the model’s parametric knowledge (e.g., "Ganesha is a Hindu god"). For instance:
> |  Question | Model output (Gemma-2-9B) |
> | - | - |
> | Kendall opened their mouth to speak and what came out shocked everyone. How would you describe Kendall? (1) a very quiet person (2) a very passive person (3) a very aggressive and talkative person | Kendall opened their mouth to speak and what came out shocked everyone. Thus, Kendall is a very **talkative** person. So the answer is: (3) a very aggressive and talkative person. |
>
> Here, we probed the token "talkative", akin to the "attribute" defined in the paper, to analyze how the model reasons about Kendall's personality based on contextual information.
>
> **Experimental Setting**:
> The data formats of WinoGrande and SocialIQA differ from the data templates used in our paper. However, both datasets contain "attributes" (factual knowledge). Consistent with the methodology presented in the paper, GPT-4 is used to locate the positions of these attributes. The experimental setup remains the same as described in the paper.

---

> ### Author Response · Authors · 2024-11-22
>
> **Part 5/6**
> --
> ***Ans to Weakness 4-part2:***
>
> **Results**:
> 1. Comparison of probing results across the three datasets (StrategyQA, WinoGrande, SocialIQA): [Probing Results](https://anonymous.4open.science/r/ecf9458a-50f9-4052-bdb8-0e79db28ed17/probing_3_dataset_on_gemma2-9b.pdf) (Consistent curve trends, e.g., attribute knowledge acquisition,  as reported in the paper.)
>
> 2. SSFT results on WinoGrande and SocialIQA: [SSFT Results](https://anonymous.4open.science/r/ecf9458a-50f9-4052-bdb8-0e79db28ed17/ssft_on_siqa_winogrande_llama.pdf) (SSFT performs better than SFT on out-of-domain datasets, while achieving comparable performance on in-domain datasets.)
>
> 3. **Explanation of the Model’s Knowledge Recall Process in the WinoGrande Case:**
> At the "vegetarian" token, shallow MLP layers engage in **knowledge augmentation**, enriching the residual stream with related knowledge (e.g., "chicken," "meat," "vegetable"). At the predicted token position, attention heads in intermediate-to-later layers transport this relevant knowledge from the residual stream at the "vegetarian" token to the current position. Probing reveals that "chicken" carries the most significant weight at this stage. However, intermediate-to-later MLP layers subsequently prioritize and incorporate the knowledge of "meat" into the residual stream. Ultimately, the MLP predicts "meat" as the output through a **re-ranking mechanism**.
>
> 4. **Explanation of the Model’s Knowledge Recall Process in the SocialIQA Case:** At the position of the predicted token, attention heads in the intermediate-to-later layers focus directly on the option tokens, transferring their associated knowledge (e.g., "quiet," "passive," "aggressive," and "talkative") to the residual stream at the predicted token position. Subsequently, the intermediate-to-later MLP layers perform **re-ranking**, prioritizing and selecting the most contextually appropriate attribute. Ultimately, the model outputs "talkative" as the final prediction.
>
> **Key Findings**:
> 1. **Consistency with Original Findings**: In both WinoGrande and SocialIQA, we observed consistent patterns as reported in the paper:
>    - The attention layers from layers 25 to 40 are responsible for retrieving relevant attributes during the Knowledge Retrieval stage.
>    - The MLP layers from layers 28 to 35 rerank these attributes to help the model output the most appropriate ones.
> 2. **Unique Observations in SocialIQA**: While Knowledge Retrieval was observed, the phenomenon of Knowledge Augmentation was less pronounced. We hypothesize this is because the required knowledge is explicitly provided in the question context, reducing the need for the model to infer additional related knowledge.
>
> ---
>
> **Question 1**
>
> > Have the authors tested the SSFT approach on larger models or more complex tasks to assess its broader applicability?
>
> ***Ans to Question 1:***
>
> We evaluated the Qwen-2.5-72B model, which is currently the top-ranked open-source model. Our evaluation was conducted on the StrategyQA dataset. Below are the results of the experiments:
>
>
> | **Models**       | **StrategyQA** | **CSQA (OOD)** | **WinoGrande (OOD)** | **SocialIQA (OOD)** | **Avg on OOD Task** |
> |-------------------|----------------|----------------|-----------------------|---------------------|---------------------|
> | **Qwen-2.5-72B**  | 86.9           | 84.1           | 78.7                 | 78.1                | 80.3                |
> | + SFT            | 90.5 (+3.6)    | 81.3 (-2.8)    | 77.3 (-1.4)          | 73.2 (-4.9)         | 77.6 (-2.7)         |
> | + SSFT           | 90.0 (+3.1)    | 86.6 (+2.5)    | 79.0 (+0.3)          | 80.0 (+1.9)         | 81.9 (+1.6)         |
>
> As shown above, SSFT methods can generalize to larger models.
>
> Additional probing results for Qwen-2.5-72B can be found in the anonymous link: [Probing Results](https://anonymous.4open.science/r/ecf9458a-50f9-4052-bdb8-0e79db28ed17/qwen72b_probing.pdf).

---

> ### Author Response · Authors · 2024-11-22
>
> **Part 6/6**
> --
> **Question 2**
>
> > How does this work differentiate itself from prior studies on retrieval-augmented generation and knowledge editing, and what unique insights does it provide on commonsense reasoning?
>
> ***Ans to Question 2:***
>
> We clarify how our work differentiates from prior studies on retrieval-augmented generation (RAG) and knowledge editing:
>
>
> 1. **Retrieval-Augmented Generation (RAG)**
>    RAG [1,2,3] treats the model as a black box and focus on augmenting its input with retrieved knowledge.
>
> 2. **Knowledge Editing (KE)**
>    KE [4,5,6,7] provides limited insights into the internal mechanisms of LLMs, as it aims to improve model performance by editing its internal knowledge.
>
> 3. **Our Work: Mechanism Interpretability**
>    In contrast, our study focuses on **mechanism interpretability**: interpreting the internal mechanisms about how LLMs perform factual knowledge recall tasks. We first locate key modules responsible for knowledge retrieval and re-ranking, then selectively improve model’s commonsense reasoning capabilities.
>
> **Unique Insights from Our Work**
>    For commonsense reasoning tasks, we identify, analyze, and interpret the internal mechanisms LLMs use to perform these tasks. We discover the formation of a **reasoning process framework** that analyzes how knowledge is augmented, retrieved, and re-ranked within the network.
>
> **References**
>
> [1] Retrieval-Augmented Generation for Knowledge-Intensive NLP Tasks. NIPS
> [2] Improving Language Models by Retrieving from Trillions of Tokens. ICML
> [3] Few-shot Learning with Retrieval Augmented Language Models. JMLR
> [4] Fast model editing at scale. ICLR
> [5] Locating and editing factual associations in GPT. NIPS
> [6] Memory-based model editing at scale. ICML
> [7] Can We Edit Factual Knowledge by In-Context Learning? EMNLP
>
> ---
>
> **Question 3**
>
> > Are there plans to extend the interpretability analysis to explore deeper relationships between reasoning components, such as hierarchical or causal connections?
>
> ***Ans to Question 3:***
>
> We have already conducted preliminary experiments to explore the circuits within different reasoning components, which bear similarities to hierarchical or causal connections.
>
> Specifically, we are performing a more detailed circuit analysis to investigate what drives the model’s re-ranking mechanism. Our approach involves using the path patching method to block pathways between shallow attention heads and the re-ranking MLP layers. This allows us to observe the information flowing into the MLP that triggers the re-ranking mechanism. In the future, we plan to further refine and expand upon this line of work.

---

> ### Author Response · Authors · 2024-11-30
>
> Dear Reviewer DWRJ,
>
> Thanks for your valuable time in reviewing and constructive comments, according to which we have tried our best to answer the questions and carefully revise the paper. Here is a summary of our response for your convenience:
>
> 1. **Limited Novelty Issue**: We compare our work with existing studies on the interpretability of factual knowledge and summarize our **insights** from three aspects: Reasoning process, Interpreting tools, and Key findings (in part 1/6).
> 2. **Marginal Improvements from SSFT**: We clarify the improvement of SSFT and reorganize the Table of SSFT results. Our SSFT results indicate that SSFT achieved significant improvements over SFT on two out-of-domain tasks while delivering comparable performance on the in-domain task (in part 2/6).
> 3. **Surface-Level Interpretability Analysis**: We analyze the model's failure in commonsense reasoning tasks by interpreting the inner mechanism within the model. The experiments reveal that the majority of errors occur during the re-ranking stage, where attention layers successfully transport relevant attributes but MLP layers select incorrect attributes for output. In addition, we provide a case study for illustration (in part 3/6).
> 4. **Narrow Experimental Scope**: We conducted additional experiments on the WinoGrande and SocialIQA datasets. Probing and SSFT results on these two datasets are consistent with our original findings (in part 4/6 & 5/6).
> 5. **Result on Larger Models**: We conduct SSFT on Qwen2.5-72B to further validate that SSFT can generalize to larger models (in part 5/6).
> 6. **Comparison between RAG and Knowledge Editing**: We compare our work with typical studies from RAG and Knowledge Editing. We claim our key insight findings compare these two studies (in part 6/6).
>
> We have revised our manuscript based on your valuable comments before the revision submission system closes. We sincerely hope our responses can address your concern. If you have any additional concerns or comments that we may have overlooked, we would greatly appreciate any further feedback from you to help us enhance our work.
>
> Best regards,
>
> Authors of #660

---

> > ### Author Response · Authors · 2024-12-04
> >
> > Dear reviewer DWRJ,
> >
> > Thank you once again for taking your time. We would be grateful if you could confirm whether our response has addressed your concerns. If you have any further questions or suggestions, please don't hesitate to let us know.
> >
> > Best regards,
> >
> > Authors of #660

---

### Official Review · Reviewer_Keqz · 2024-11-04

**Soundness:** 3
**Presentation:** 3
**Contribution:** 3
**Rating:** 6
**Confidence:** 4

**Summary:**

This paper investigates the commonsense reasoning capabilities of large language models (LLMs). The authors aim to make the reasoning process of LLMs more transparent and understandable by dissecting it into human-comprehensible steps. Through the analysis of hidden states in different transformer layers and token positions, the paper uncovers a mechanism by which LLMs execute reasoning. The commonsense reasoning process in LLMs is found to involve a sequence of knowledge augmentation, retrieval, and answer generation, similar to retrieval-augmented generation. The paper also identifies that LLMs often contain relevant factual knowledge but fail to retrieve it correctly. To address this, the authors selectively fine-tune key heads and MLPs, leading to improvements in reasoning performance.

**Strengths:**

-	The proposed approach provides a understanding of the inner workings of LLMs during commonsense reasoning.
-	The paper introduces a selective fine-tuning strategy that targets less than 10% of the model's parameters, leading to notable performance enhancements. It is particularly effective for out-of-domain settings.

**Weaknesses:**

-	The experiments are mainly conducted on specific models like Llama2-7B and Gemma2-9B. It would be better to expand the coverage of the experiments.
-	Although GPT-4 is used to assist in the analysis, there may still be some uncertainty in the interpretation of the results. The "Interpreting Module" may not be completely accurate in all cases.

**Questions:**

1.	How do the authors ensure that the selective fine-tuning approach does not lead to overfitting, particularly when focusing on specific components of the model?

---

> ### Author Response · Authors · 2024-11-22
>
> **Part 1/1**
> --
> **Weakness 1**
> > The experiments are mainly conducted on specific models like Llama2-7B and Gemma2-9B. It would be better to expand the coverage of the experiments.
>
> ***Ans to Weakness 1:***
>
> We evaluated the Qwen-2.5-72B model, which is currently the top-ranked open-source model. Our evaluation was conducted on the StrategyQA dataset. Below are the results of the experiments:
>
>
> | **Models** | **StrategyQA** | **CSQA (OOD)** | **WinoGrande (OOD)** | **SocialIQA (OOD)** | **Avg on OOD Task** |
> |-|-|-|-|-|-|
> | **Qwen-2.5-72B**  | 86.9  | 84.1   | 78.7    | 78.1   | 80.3   |
> | + SFT   | 90.5 (+3.6)  | 81.3 (-2.8)    | 77.3 (-1.4)   | 73.2 (-4.9)  | 77.6 (-2.7) |
> | + SSFT  | 90.0 (+3.1)   | 86.6 (+2.5)    | 79.0 (+0.3)  | 80.0 (+1.9) | 81.9 (+1.6)  |
>
> As shown above, SSFT performs better than SFT on out-of-domain datasets, while achieving comparable performance on in-domain datasets, which indicates our SSFT methods can generalize to larger models.
>
> Additional probing results for Qwen-2.5-72B can be found in the anonymous link: [Probing Results](https://anonymous.4open.science/r/ecf9458a-50f9-4052-bdb8-0e79db28ed17/qwen72b_probing.pdf). (If the image in the link does not load, refreshing or downloading the link could resolve the issue.)
>
> ---
>
> **Weakness 2**
>
> > Although GPT-4 is used to assist in the analysis, there may still be some uncertainty in the interpretation of the results. The "Interpreting Module" may not be completely accurate in all cases.
>
> ***Ans to Weakness 2:***
>
> We conduct experiments to compare the results of "GPT-4" and "human validation".
> In the paper, GPT-4 is applied in:
> - Generation of $X_c$ (counterfactual data) using GPT-4 ;
> - Generation of the analysis of key component behavior using GPT-4;
> - Generation of data template;
>
> For all the scenarios, we engaged ten master's students specializing in Natural Language Processing as volunteers. Five students were tasked with executing all procedures manually, including generating $X_c$, analyzing key component behaviors, and developing data templates. The remaining students then compared their annotations with those generated by GPT-4 to judge which more accurately represented the component behavior. The results of this evaluation are shown in the table below:
> | Scenarios   | GPT Wins | Human Wins | Ties |
> |-|-|-|-|
> | Generation of $X_c$ using GPT-4     | 8%  | 12%   | 80%  |
> | Analysis of key component behavior using GPT-4      | 12%      | 10%    | 78%  |
> | Generation of data template        | 7%    | 18%  | 75%  |
>
> Overall, the results demonstrate that GPT-4 is highly accepted by human evaluators, with the combination of "GPT wins" and "Ties" exceeding 80%, underscoring its robust reliability. These indicate that **GPT-4's outputs are almost consistent with those generated by humans.**
>
> Throughout the analysis process, GPT-4 played a key role in quickly summarizing and synthesizing information. In practice, we manually spot-checked and validated GPT-4's outputs to ensure their accuracy and reliability. Once intermediate results fell short of expectations, human intervention was introduced to refine and re-execute the process. In our experiments, the proportion of cases requiring human intervention closely mirrored the proportions of "Human wins". Essentially, GPT-4 primarily served the role of rapid summarization and pattern identification, allowing us to analyze a larger number of cases and uncover broader, generalizable reasoning patterns.
>
> ---
> **Question 1**
> > How do the authors ensure that the selective fine-tuning approach does not lead to overfitting, particularly when focusing on specific components of the model?
>
> ***Ans to Question 1:***
>
> Thank you for your question. We have taken several measures to ensure that the selective fine-tuning (SSFT) approach does not lead to overfitting, particularly when targeting specific components of the model:
>
> 1. **Generalization Across Datasets**:
>    After fine-tuning on the multi-hop commonsense reasoning dataset **StrategyQA** (using the 7B model as an example), we evaluated the model on other commonsense reasoning datasets (e.g., CSQA) and a math reasoning dataset with a larger domain gap (e.g., GSM8K). The results show an accuracy improvement of **+3.0%** on CSQA and comparable performance on GSM8K (**-0.1%**). These results suggest that overfitting did not occur.
>
> 2. **Hypothesis on Generalization**:
>    We hypothesize that the observed improvement on other commonsense reasoning datasets is due to the targeted enhancement of the model’s knowledge recall ability via selective fine-tuning. Additionally, the unchanged performance on a domain-gap dataset like GSM8K likely reflects the minimal parameter adjustments made during SSFT (only **3.0%** of parameters were updated), reducing the risk of overfitting.
>
> We hope this clarifies our approach and the rationale behind ensuring robust generalization while avoiding overfitting. Thank you for highlighting this important aspect.

---

### Official Review · Reviewer_scHF · 2024-11-04

**Soundness:** 3
**Presentation:** 3
**Contribution:** 3
**Rating:** 5
**Confidence:** 4

**Summary:**

This paper investigates how Large Language Models (LLMs) perform commonsense reasoning by deciphering their internal mechanisms. The researchers discovered that LLMs execute reasoning through a five-stage process similar to Retrieval-Augmented Generation (RAG): knowledge augmentation, recall, re-ranking, rationale conclusion, and answer generation. The study found that both attention heads and multi-layer perceptrons (MLPs) contribute to factual knowledge generation from different perspectives. When LLMs fail at commonsense reasoning tasks, it's often due to incorrect knowledge retrieval rather than a lack of knowledge. To address this, the researchers developed a selective fine-tuning approach targeting specific attention heads and MLPs crucial for knowledge retrieval, which improved reasoning performance while modifying less than 10% of the model's parameters. The study was conducted using Llama2 and Gemma models, focusing on a standardized template for commonsense reasoning questions to ensure controlled experimentation.

**Strengths:**

1. This paper studies the internal mechanism for LLMs to conduct commonsense reasoning. The authors split the reasoning processes into 5 stages and find that the retrieval stage is the main source of reasoning errors. This paper provides insightful discovery of the underlying mechanism of LLMs.
2. Based on the new discovery, this paper also proposed a fine-tuning method that only focuses on top-K attention heads.
3. Extensively experimental results show the effectiveness of the methods.

**Weaknesses:**

1. The evaluation is limited to a small scope. The authors conduct their experiments solely on yes or no questions in the commonsense domain. More datasets with diverse formats should be included, like WinoGrande and SocialIQA. This eliminated scope impairs the conclusion of this paper and raises the doubt of generalization in more formats and more domains.
2. The evaluation of the paper heavily relies on GPT-4 for both data synthesis and analysis verification. The accuracy of GPT-4 on those tasks remains unclear, and the authors need to provide more experiments to show the agreement between GPT-4 and human experts.
3. The authors just used others' methods for efficient fine-tuning without any modification. In Section 3.3, the authors only use a single paragraph to finish the description of their methods for efficient tuning. Meanwhile, this methods is just a copy of previous method from the paper "Interpreting and improving large language models in arithmetic calculation."

**Questions:**

No

---

> ### Author Response · Authors · 2024-11-22
>
> **Part 1/3**
> --
> **Weakness 1**
> > The evaluation is limited to a small scope. The authors conduct their experiments solely on yes or no questions in the commonsense domain. More datasets with diverse formats should be included, like WinoGrande and SocialIQA. This eliminated scope impairs the conclusion of this paper and raises the doubt of generalization in more formats and more domains.
>
> ***Ans to Weakness 1-part1:***
>
> We conducted additional experiments on the WinoGrande and SocialIQA datasets. These datasets provide complementary perspectives as they differ significantly in format from the dataset used in our paper. For these two datasets, we adopted a multi-choice format so as to prove that our analysis method can generalize to more domains. We provide a detailed overview and features of these datasets.
>
> **WinoGrande**:
> The factual knowledge required by WinoGrande is relatively straightforward compared to the Wikipedia-based knowledge in StrategyQA. For example:
> |  Question | Model output (Gemma-2-9B) |
> | - | - |
> | It was easy for Angela to become a vegetarian although Kayla couldn't do it. _ really missed the taste of chicken. (1) Angela (2) Kayla  |  A person is a vegetarian means they do not eat **meat**. A person who really misses the taste of chicken likely enjoys eating chicken. Since Angela became a vegetarian and Kayla could not, Kayla really missed the taste of chicken. |
>
> In this example, we probed the token "**meat**", akin to the "attribute" in factual knowledge defined in the paper, to investigate how the model encodes factual knowledge such as "a vegetarian does not eat meat."
>
> **SocialIQA**:
> In SocialIQA, the reasoning is based on knowledge provided in the context, whereas our original experiments relied on the model’s parametric knowledge (e.g., "Ganesha is a Hindu god"). For instance:
> |  Question | Model output (Gemma-2-9B) |
> |-|-|
> | Kendall opened their mouth to speak and what came out shocked everyone. How would you describe Kendall? (1) a very quiet person (2) a very passive person (3) a very aggressive and talkative person | Kendall opened their mouth to speak and what came out shocked everyone. Thus, Kendall is a very **talkative** person. So the answer is: (3) a very aggressive and talkative person. |
>
> Here, we probed the token "talkative", akin to the "attribute" defined in the paper, to analyze how the model reasons about Kendall's personality based on contextual information.
>
> **Experimental Setting**:
> The data formats of WinoGrande and SocialIQA differ from the data templates used in our paper. However, both datasets contain "attributes" (factual knowledge). Consistent with the methodology and the process of the experiment presented in the paper, GPT-4 is used to locate the positions of these attributes. The experimental setup remains the same as described in the paper.
>
> **Results**:
> (Results are provided via anonymous links as OpenReview does not support image uploads. If the image in the link does not load, refreshing or downloading could resolve the issue.)
> 1. Comparison of probing results across the three datasets (StrategyQA, WinoGrande, SocialIQA): [Probing Results](https://anonymous.4open.science/r/ecf9458a-50f9-4052-bdb8-0e79db28ed17/probing_3_dataset_on_gemma2-9b.pdf) (Consistent curve trends in knowledge retrieval as reported in the paper.)
>
> 2. SSFT results on WinoGrande and SocialIQA: [SSFT Results](https://anonymous.4open.science/r/ecf9458a-50f9-4052-bdb8-0e79db28ed17/ssft_on_siqa_winogrande_llama.pdf) (SSFT performs better than SFT on out-of-domain datasets while achieving comparable performance on in-domain datasets.)
>
> 3. **Explanation of the Model’s Knowledge Recall Process in the WinoGrande Case:**
> At the "vegetarian" token, shallow MLP layers engage in **knowledge augmentation**, enriching the residual stream with related knowledge (e.g., "chicken," "meat," "vegetable"). At the predicted token position, attention heads in intermediate-to-later layers transport this relevant knowledge from the residual stream at the "vegetarian" token to the current position (**knowledge retrieval**). Probing reveals that "chicken" carries the most significant weight at this stage. However, deep MLP layers subsequently prioritize and incorporate the knowledge of "meat" into the residual stream. Ultimately, the MLP predicts "meat" as the output through a **re-ranking mechanism**.
>
> 4. **Explanation of the Model’s Knowledge Recall Process in the SocialIQA Case:** At the position of the predicted token, attention heads in the intermediate-to-later layers focus directly on the option tokens, transferring their associated knowledge (e.g., "quiet," "passive," "aggressive," and "talkative") to the residual stream at the predicted token position (**knowledge retrieval**). Subsequently, the intermediate-to-later MLP layers perform **re-ranking**, prioritizing and selecting the most contextually appropriate attribute. Ultimately, the model outputs "talkative" as the final prediction.

---

> ### Author Response · Authors · 2024-11-22
>
> **Part 2/3**
> --
> ***Ans to Weakness 1-part2:***
>
> **Key Findings**:
> 1. **Consistency with Original Findings**: In both WinoGrande and SocialIQA, we observed consistent patterns as reported in the paper:
>    - The attention layers from layers 25 to 40 are responsible for retrieving relevant attributes during the Knowledge Retrieval stage.
>    - The MLP layers from layers 28 to 35 rerank these attributes to help the model output the most appropriate ones.
> 2. **Unique Observations in SocialIQA**: While Knowledge Retrieval was observed, the phenomenon of Knowledge Augmentation was less pronounced. We hypothesize this is because the required knowledge is explicitly provided in the question context, reducing the need for the model to infer additional related knowledge.
>
> ---
>
> **Weakness 2**
> > The evaluation of the paper heavily relies on GPT-4 for both data synthesis and analysis verification. The accuracy of GPT-4 on those tasks remains unclear, and the authors need to provide more experiments to show the agreement between GPT-4 and human experts.
>
> ***Ans to Weakness 2:***
>
> We conduct experiments to compare the results of "GPT-4" and "human validation".
> In the paper, GPT-4 is applied in:
> - Generation of $X_c$ (counterfactual data) using GPT-4 ;
> - Generation of the analysis of key component behavior using GPT-4;
> - Generation of data template;
>
> For all the scenarios, we engaged ten master's students specializing in Natural Language Processing as volunteers. Five students were tasked with executing all procedures manually, including generating $X_c$, analyzing key component behaviors, and developing data templates. The remaining students then compared their annotations with those generated by GPT-4 to judge which more accurately represented the component behavior. The results of this evaluation are shown in the table below:
> | Scenarios                                           | GPT Wins | Human Wins | Ties |
> |-----------------------------------------------------|----------|------------|------|
> | Generation of $X_c$ using GPT-4                     | 8%       | 12%        | 80%  |
> | Analysis of key component behavior using GPT-4      | 12%      | 10%        | 78%  |
> | Generation of data template                         | 7%       | 18%        | 75%  |
>
> Overall, the results demonstrate that GPT-4 is highly accepted by human evaluators, with the combination of "GPT wins" and "Ties" exceeding 80%, underscoring its robust reliability. These indicate that **GPT-4's outputs are almost consistent with those generated by humans.**
>
> Throughout the analysis process, GPT-4 played a key role in quickly summarizing and synthesizing information. In practice, we manually spot-checked and validated GPT-4's outputs to ensure their accuracy and reliability. Once intermediate results fell short of expectations, human intervention was introduced to refine and re-execute the process. In our experiments, the proportion of cases requiring human intervention closely mirrored the proportions of "Human wins". Essentially, GPT-4 primarily served the role of rapid summarization and pattern identification, allowing us to analyze a larger number of cases and uncover broader, generalizable reasoning patterns.

---

> ### Author Response · Authors · 2024-11-22
>
> **Part 3/3**
> --
> **Weakness 3**
>
> > The authors just used others' methods for efficient fine-tuning without any modification. In Section 3.3, the authors only use a single paragraph to finish the description of their methods for efficient tuning. Meanwhile, this method is just a copy of the previous method from the paper "Interpreting and improving large language models in arithmetic calculation."
>
> ***Ans to Weakness 3:***
>
> Actually, we seek to validate the **effectiveness and importance of the interpreted mechanism, where the existing selective fine-tuning approach is appropriate to serve this purpose**.  Moreover, avoiding modifications to the existing methods allows us to more accurately underscore the effectiveness of the identified modules for factual knowledge recall.
>
> To address your concerns, we conducted additional experiments with alternative validation methods, including:
>
> 1. **Combining SSFT with LoRA**:
>   We applied LoRA (with \( r=64 \) and \( \alpha=32 \)) to the outputs of the identified modules, combining sparse selective supervised fine-tuning (SSFT) with LoRA. This approach enabled us to investigate the impact of low-rank adaptations on key modules. Our findings show that this method achieves an additional 3–4% average accuracy improvement on out-of-domain (OOD) datasets (compared to SFT) while improving accuracy by 20% on in-domain (ID) datasets such as WinoGrande (comparable to SFT).
>
> 3. **Representation Control**:
>    Following [1], we performed representation control to adjust model behavior by intervening in the outputs of the identified MLPs for **Re-ranking**. For example:
>
> |  Question | Model output (Gemma-2-9B) | Model output after control |
> | -------- | -------- | -------- |
> | Q: Yes or no: Would Persephone be a good consultant to a landscape architect?  | Persephone is the Greek goddess of the **underworld**. Landscape architects design gardens. Thus, Persephone would not be a good consultant to a landscape architect. So the answer is no.  | Persephone is the Greek goddess of **spring**. Thus, Persephone would be a good consultant to a landscape architect. So the answer is yes.  |
>
>
>    Here, the original response reaches an incorrect conclusion using the factual knowledge "Persephone is the Greek goddess of the **underworld**." Though correct, the context suggests a more relevant recall of the attribute "Persephone is the Greek goddess of the **spring**." We utilized Sparse Autoencoder (SAE) to identify the features related to "spring" within the MLP layer for re-ranking. By increasing the weight of the "spring" feature in the MLP outputs, we were able to control the model’s prediction, shifting it from "underground" to "spring." Experimental results revealed that only controlling the MLP outputs responsible for re-ranking effectively altered the model's behavior, while controlling other MLP layers had almost no impact. Ultimately, we successfully resolved 73% of the failure cases on the StrategyQA dataset.
>
>
>
> **Case Study: Mechanism Validation**:
> We conducted an interpretability analysis on a specific case to examine changes in the model before and after fine-tuning. The results are available at the following anonymous link: [Mechanism Validation](https://anonymous.4open.science/r/ecf9458a-50f9-4052-bdb8-0e79db28ed17/case_study_on_sft_base.pdf).
>
> Our findings are as follows:
> 1. After training, the model’s **attention layers** demonstrated a significant improvement in their ability to retrieve correct knowledge. Compared to the base model, the information related to the incorrect reasoning token ("shorter") in the 26th attention layer output was significantly reduced in the trained model.
> 2. After training, the model’s **MLP layers** exhibited a marked enhancement in re-ranking capability. Compared to the base model, the information related to the correct reasoning token ("longer") in the 26th MLP layer output was significantly increased in the trained model.
> 3. Comparing the results before and after training the Head and MLP modules, we observed significant improvements in the model’s knowledge retrieval and re-ranking accuracy within the targeted modules. This validates the effectiveness and reliability of the interpreted mechanism.
>
>
> **References**:
> [1] Scaling and Evaluating Sparse Autoencoders. OpenAI 2024

---

> > ### Comment · Reviewer_scHF · 2024-11-26
> > **Response to authors**
> >
> > Thanks for the clarification and the large amount of new experiments. I raised my score.

---

> ### Author Response · Authors · 2024-11-27
>
> Dear Reviewer scHF,
>
> We are delighted to hear that the additional experiments we included addressed your concerns. We sincerely appreciate your decision to raise the score. If you have any further questions or suggestions regarding our work or methodology, we would be more than happy to engage in further discussion.
>
> Thank you once again for taking the time to review our paper!
>
> Best regards,
>
> The Authors of Paper #660

---

### Author Response · Authors · 2024-11-26
**General Response About Revision**

First of all, we would like to express our gratitude to all the reviewers for their time and efforts in reviewing our paper. The valuable comments from the reviewers have greatly contributed to strengthening our manuscript.

We have revised our manuscript based on the reviewers' comments and have uploaded it. Here, we summarize our modifications, following the original order presented on this webpage:


1. We have added a comparison with related work on factual knowledge interpretability (Thanks to Reviewer DWRJ).

2. We have included formal definitions for the key concepts used in the paper in Section 3.1 (Preliminary) and adjusted the terminology used in the experimental sections based on these definitions (Thanks to Reviewer 4kUU).

3. We have reorganized the explanation of the interpreting module in Section 3.2, refining its framework and process (Thanks to Reviewer 4kUU).

4. We have added Section 3.3 (Instantiation of Interpreting Module) to provide specific details on how the interpreting module is applied. Additionally, we have included a diagram in the appendix to illustrate the input and output of each interpreting module (Thanks to Reviewer 4kUU).

5. We have reviewed the parts of the paper involving GPT and conducted a systematic evaluation of the performance differences between GPT and humans, providing the details of the prompts used (Thanks to Reviewers keqz, scHF).

6. We have added Section 4.1 (Experiment Overview) at the beginning of the experimental section to clarify the structure of the experiments and the relationships between the sections (Thanks to Reviewer 4kUU).

7. We have included probing experiment results for Gemma2-9B on two new datasets (Winogrande and SocialIQA) and validated the effectiveness of SSFT on these new datasets (Thanks to Reviewers scHF, DWRJ).

8. We have added probing and SSFT experimental results for Qwen2.5-72B on StrategyQA (Thanks to Reviewers keqz, DWRJ, 4kUU).

9. We have included an explanation of the interpretability analysis before and after SSFT training, demonstrating how SSFT enhances the model's capabilities (Thanks to Reviewer scHF).

10. We have reorganized the experimental tables for improved clarity and coherence (Thanks to Reviewer DWRJ).

11. We have updated the captions for all figures in the experimental sections, providing more detailed explanations to help readers understand the key ideas presented in each figure (Thanks to Reviewer 4kUU).

12. We have corrected errors in the figure from Section 4.6, issues and citation format errors (Thanks to Reviewer 4kUU).

Finally, we would like to thank all the reviewers once again for taking the time to review our paper and providing us with your invaluable feedback. We sincerely appreciate your constructive comments on our work.

Please feel free to ask any remaining questions as we would be glad to provide further clarification. We look forward to any further discussions with all the reviewers.

---

### Meta-Review · Area_Chair_TyVt · 2024-12-25

**Metareview:**

This paper investigates the internal commonsense reasoning mechanisms in large language models (LLMs) by interpreting hidden states across transformer layers and token positions. The authors identify a five-stage reasoning process involving knowledge augmentation, retrieval, re-ranking, rationale conclusion, and answer generation, akin to retrieval-augmented generation. To address common retrieval errors, they introduce a (parameter-efficient) Selective Supervised Fine-Tuning (SSFT) approach, targeting specific attention heads and MLP layers, which improves reasoning performance.

Strength: The paper provides a detailed analysis of LLMs’ internal mechanisms during commonsense reasoning, offering insights into key components like attention heads and MLP layers. The proposed resource-efficient SSFT method demonstrates performance improvements.

Weakness: Reviewers pointed out that the evaluation scope is narrow, with experiments focusing on limited datasets and specific models. This has been partially addressed during the rebuttal. Also the novelty of the work is limited to applying existing techniques to analyzing a new problem.

**Additional Comments On Reviewer Discussion:**

One major concern raised by most of the reviewers is the limited scope of evaluation on limited datasets and models. The authors have addressed this by adding new experiments, which (partially) addressed the feedback from the reviewers. However, fully addressing the reviews requires a substantial revision of the paper, as indicated in so many blue texts in the revised draft. Therefore, given the substantial revision, this paper needs a new round of fresh review before it can be accepted. Nevertheless, this paper contains quite meaningful contribution to the area. The authors are encouraged to more thoroughly revise the paper based on the feedback and submit it to a future venue.

---

### Decision · Program_Chairs · 2025-01-22

Reject